# Direct observation of kink evolution due to Hund's coupling on approach to metal-insulator transition in $NiS_{2-x}Se_x$

Bo Gyu Jang [1,2,13], Garam Han [3,4,13], Ina Park[1], Dongwook Kim[1], Yoon Young Koh[5], Yeongkwan Kim[6], Wonshik Kyung [3,4], Hyeong-Do Kim[3,4,12], Cheng-Maw Cheng[7], Ku-Ding Tsuei[7], Kyung Dong Lee[8], Namjung Hur[8], Ji Hoon Shim [1,9,10 ✉], Changyoung Kim [3,4 ✉] & Gabriel Kotliar[11]

Understanding characteristic energy scales is a fundamentally important issue in the study of strongly correlated systems. In multiband systems, an energy scale is affected not only by the effective Coulomb interaction but also by the Hund's coupling. Direct observation of such energy scale has been elusive so far in spite of extensive studies. Here, we report the observation of a kink structure in the low energy dispersion of $NiS_{2-x}Se_x$ and its characteristic evolution with $x$, by using angle resolved photoemission spectroscopy. Dynamical mean field theory calculation combined with density functional theory confirms that this kink originates from Hund's coupling. We find that the abrupt deviation from the Fermi liquid behavior in the electron self-energy results in the kink feature at low energy scale and that the kink is directly related to the coherence-incoherence crossover temperature scale. Our results mark the direct observation of the evolution of the characteristic temperature scale via kink features in the spectral function, which is the hallmark of Hund's physics in the multiorbital system.

[1] Department of Chemistry, Pohang University of Science and Technology, Pohang, Korea. [2] Center for High Pressure Science and Technology Advanced Research, Shanghai, China. [3] Center for Correlated Electron Systems, Institute for Basic Science (IBS), Seoul, Korea. [4] Department of Physics and Astronomy, Seoul National University, Seoul, Korea. [5] Max Plank POSTECH Center for Complex Phase Materials, Pohang University of Science and Technology, Pohang, Korea. [6] Department of Physics, KAIST, Daejeon, Korea. [7] National Synchrotron Radiation Research Center, Hsinchu, Taiwan. [8] Department of Physics, Inha University, Incheon, Republic of Korea. [9] Department of Physics, Pohang University of Science and Technology, Pohang, Korea. [10] Division of Advanced Materials Science, Pohang University of Science and Technology, Pohang, Korea. [11] Department of Physics and Astronomy, Rutgers University, Piscataway, NJ, USA. [12] Present address: PAL-XFEL, Pohang Accelerator Laboratory, Pohang, Korea. [13] These authors contributed equally: Bo Gyu Jang, Garam Han. ✉email: jhshim@postech.ac.kr; changyoung@snu.ac.kr

Since the Mott's initial proposal that an insulating ground state can appear due to the electron–electron correlation, the metal-insulator transition (MIT) has been at the core of condensed matter physics. The Coulomb interaction $U$ is the most important parameter, and thus finding how the spectral function and energy scale evolve as a function of $U$ has been a fundamental issue in the MIT studies. The Brinkman-Rice picture and dynamical mean-field theory (DMFT) for half-filled one-band Hubbard model show that the overall quasi-particle (QP) peak and Kondo temperature $T_K$ gradually become renormalized as the $U$ increases. At the MIT, the QP mass diverges with vanishing $T_K$.

Most of realistic materials are, however, multiorbital systems in which not only $U$ but also the Hund's coupling $J_H$ is a critical parameter for the ground state. During the last decade, there has been a remarkable progress in the theoretical description of the Hund's physics in correlated electron systems. It was found that $J_H$ can enhance the effective correlation strength of multiorbital systems by weakening the Kondo screening channel[1–4]. The most drastic effect occurs in non-singly-occupied and non-half-filled cases such as iron pnictides, chalcogenides and ruthenates[1,2,5–7]. Although these materials are metallic and are located far away from the Mott insulating state, their small coherence energy scale due to $J_H$ induces incoherent transport properties. These new phases are classified into Hund's metal and their correlated electronic structures have been intensively studied through both experimental and theoretical approaches.

An important remaining question is how $J_H$ affects the evolution of the spectral function and the energy scale of multiband systems. Unfortunately, varying the correlation strength over a wide interaction range in Hund's metal systems is found to be difficult. Due to the Janus-faced character of $J_H$ in non-singly-occupied and non-half-filled cases, it is hard to reduce the QP weight $Z$ further below a certain point, where most of realistic Hund's metals are located[1,2,8]. Although the Hund's physics has been studied one unit away from the half-filled case, the same low-energy effect is expected for the half-filled case in the presence of $J_H$[1–3,8]. Considering these aspects, NiS$_{2-x}$Se$_x$, a half-filled system with degenerate Ni $e_g$ orbitals, is probably the most

suitable multiorbital system for an investigation of the evolution in the presence of $J_H$. NiS$_2$ ($x = 0.0$) is a well-known Mott insulator and goes through a bandwidth controlled MIT without any structural change to a correlated metal NiSe$_2$ ($x = 2.0$) at $x = 0.5$. By varying the Se content, the correlation strength can be easily tuned in the existence of $J_H$. Therefore, NiS$_{2-x}$Se$_x$ provides an interesting playground to study the Hund's physics in the half-filled case near the MIT, unlike the reported Hund's metal system.

The correlated electronic structures of NiS$_{2-x}$Se$_x$ have been indeed studied for a long time, not only theoretically by density functional theory plus dynamical mean-field theory (DFT+DMFT) calculations[9,10] but also experimentally by angle-resolved photoemission spectroscopy (ARPES)[11–13]. However, the observation of the QP behavior in previous ARPES studies was rather elusive; the QP sits on an incoherent background and the QP information could not be clearly obtained in a direct fashion. In order to address the role of $J_H$ during the MIT, we reexamine the band structure of NiS$_{2-x}$Se$_x$ not only by ARPES with finer doping steps and higher resolution but also via DFT+DMFT with and without $J_H$. We utilize a low photon energy ($h\nu = 60$–63 eV and 100–106 eV) ARPES to achieve a high resolution needed to clearly observe the QP of NiS$_{2-x}$Se$_x$. Our ARPES results reveal clear QP dispersions as well as doping dependent low-energy kink structures. DFT+DMFT calculations also identify the kink structures, which explains the strongly suppressed temperature scale due to $J_H$. The evolution of kink observed in our ARPES data is the direct spectroscopic evidence for the evolution of the energy scale in the presence of $J_H$.

## Results

**Electronic structure of NiSe$_2$.** We first look at the metallic end composition NiSe$_2$ ($x = 2.0$) for which the band structure is the clearest from ARPES measurement and DFT+DMFT calculations. For DFT+DMFT calculations, the optimized parameters ($U = 5.7$ eV, $J_H = 0.7$ eV) that can describe the experimental phase diagram along both $x$ and $T$ axes are used[10]. Figure 1a, b shows the ARPES Fermi surface (FS) map near the $k_z = 0$ plane obtained by using 100 eV photon and 60 eV photon, and they are

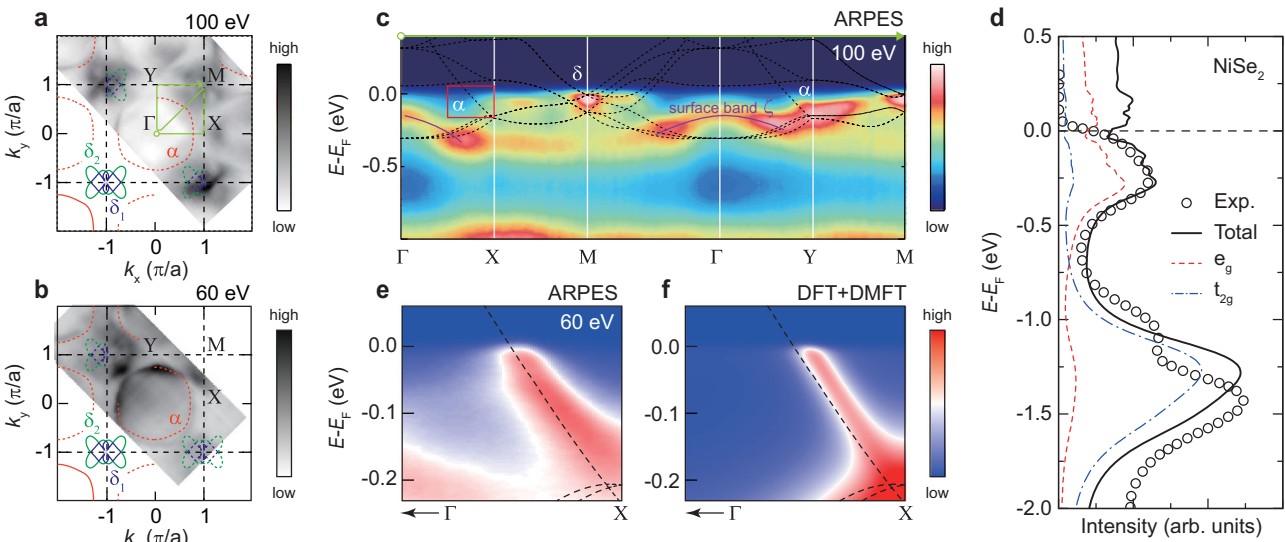

**Fig. 1 Electronic structure of NiSe$_2$. a**, **b** Fermi surface (FS) maps from ARPES measurement obtained by using 100 and 60 eV photon, respectively. The dashed lines indicate the calculated FS using DFT. **c** Band structure along the green line in panel (**a**) from ARPES. The dashed lines in **c** indicate the renormalized DFT band structure. Except for hole-like surface band $\zeta$, complex $\delta$ bands near M point and $\alpha$ band dispersion near Y point are well described by the renormalized DFT band structure. **d** $k$-integrated spectral function from ARPES and DFT+DMFT calculation. **e**, **f** $\alpha$ band near X point from ARPES and DFT+DMFT calculation, respectively. They are from the red boxed region in (**c**).

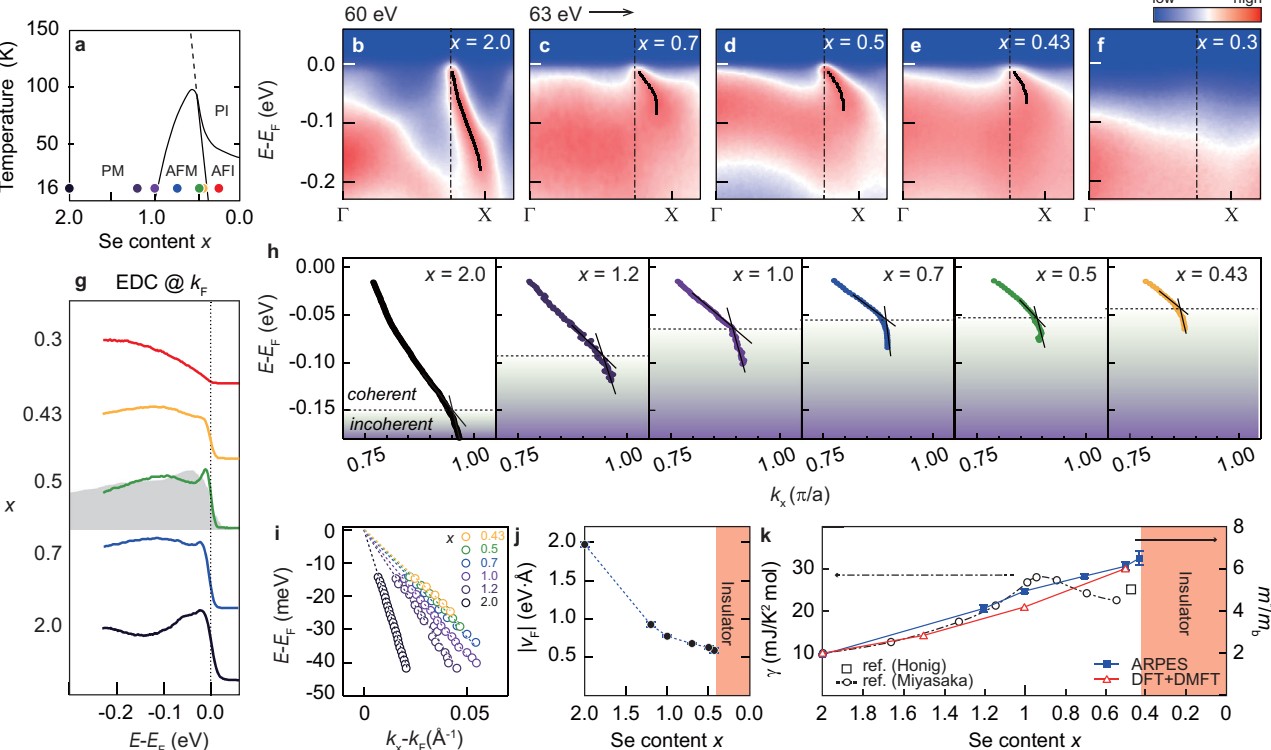

**Fig. 2 Se content-dependent quasiparticle (QP) dispersion. a** Phase diagram of NiS$_{2-x}$Se$_x$ reproduced from ref. [15]. PM, PI, AFM, and AFI stand for paramagnetic metal, paramagnetic insulator, antiferromagnetic metal, and antiferromagnetic insulator, respectively. Small circles indicate where ARPES measurements were carried out. All data were taken at 16 K. **b–f** ARPES data along the Γ-X direction (see Supplementary Fig. 6 for $x = 1.2$ and 1.0 data). **g** Energy distribution curves (EDCs) at the Fermi momentum ($k_F$) as represented by dash-dotted lines in panel (**b–f**). The QPs of previous APRES data was buried in incoherent band as depicted with gray filled curve[12]. In case of $x = 0.3$ data (Mott insulator), the EDC is from the same momentum as that of others. **h** QP dispersions obtained by fitting the momentum distribution curves (MDCs). **i–k** Doping dependent α band dispersion, Fermi velocity $v_F$, and effective mass $m^\star$, respectively. The linear coefficient of the specific heat $\gamma$ is also plotted for comparison[15,16].

consistent with the calculated FS in the $k_z = 0$ plane using DFT (dashed lines in Fig. 1a, b). One can easily see a large hole pocket α centered at the Γ point and $\delta_{1,2}$ bands near the M point. These features can also be identified in the ARPES data along the high symmetry cuts (along the green line in Fig. 1a) as shown in Fig. 1c. Other than the hole-like surface band ζ observed around Γ point in ARPES data (see Supplementary Note 2), the DFT band structure of Ni $e_g$ orbitals (dotted lines in Fig. 1c) renormalized by the factor 1.99 is consistent with ARPES data near the Fermi level. The renormalized factor, mass enhancement, is estimated from the DMFT self-energy ($m^*/m = Z^{-1} = 1 - \partial \text{Im}\Sigma(i\omega)/\partial\omega|_{\omega \to 0^+}$). One can see the α band dispersion along Γ–X and Γ–Y cuts, and the complex δ bands near M point in the experimental data. The $k$-integrated spectral function obtained from DFT+DMFT calculation agrees well with the experimental result as shown in Fig. 1d.

The α hole pocket is the most representative QP band for the Mott transition in NiS$_{2-x}$Se$_x$. Since the FS volume of the α band is much larger than the others, the transport properties of NiS$_{2-x}$Se$_x$ should be dominated by the hole pocket[14]. In order to study the α band dispersion, 60 eV photon is utilized for another $k_z = 0$ plane but for the higher resolution. Although the data taken with 100 eV may better represent the overall spectral function, the α band dispersion is more clearly seen with 60 eV photon (see Supplementary Note 3). Figure 1e, f shows the the ARPES and DFT+DMFT results, respectively, of the NiSe$_2$ α band from the red boxed region in Fig. 1c. The consistency between experiment and theory has an added benefit that DFT+DMFT calculations can be used to understand the electronic correlations.

**α band dispersion as a function of Se content**. To study how the α band dispersion varies across the MIT, ARPES spectra along the Γ-X line were taken at 16 K for a wide range of Se doping shown in Fig. 2a, and are plotted in Fig. 2b–f. Our ARPES and resistivity (Supplementary Note 1) data are consistent with the well-known phase diagram of NiS$_{2-x}$Se$_x$[15]. A QP band, distinct from the incoherent band (Fig. 2g), is clearly observed in all metallic samples, while the QP was not clearly discernible as it was buried under incoherent spectral weight in previous reports (gray filled curve in Fig. 2g)[11–13]. Appearance/disappearance of the QP follows the MIT behavior along the Se doping; the QP is seen for metallic phase ($x \geq 0.43$) while it is absent in the insulating phase ($x = 0.3$) (see Fig. 2a for the phase diagram). This also holds true along the temperature axis (Supplementary Note 4). The α band is of bulk origin as confirmed by photon-energy dependence (Supplementary Note 3) as well as bulk sensitive soft X-ray ARPES[13].

The black solid lines in Fig. 2b–f are the QP dispersions obtained by fitting the momentum distribution curves (MDCs) (Supplementary Note 5). While the Fermi momentum $k_F$ remains the same at 0.4 Å$^{-1}$ as denoted by the black dash-dot lines, it is seen that the slope of the band near the Fermi level decreases. The dispersions are redrawn in Fig. 2h to see the evolution of the dispersions more clearly. As our ARPES data clearly show the QP bands, the near $E_F$ dispersions depicted in Fig. 2i can be used to obtain the Fermi velocity $v_{F_{ARPES}}$ to investigate the mass enhancement in NiS$_{2-x}$Se$_x$. So obtained $v_{F_{ARPES}}$ is plotted in Fig. 2j. Note that $v_{F_{ARPES}}$ is finite approaching the MIT, revealing

that the effective mass $m^*$ remains finite at the transition. The mass enhancement $m^*/m$ obtained by using the relation $m^*/m = v_{F_{DFT}}/v_{F_{ARPES}}$ is plotted in Fig. 2k along with $m^*/m$ calculated from DMFT self-energy based on paramagnetic calculation at 50 K. The mass enhancement $m^*/m$ from DMFT is found to be in good agreement with that from experiment, with both remaining finite approaching the MIT. This is consistent with the behavior of the Sommerfeld coefficient $\gamma$ shown in the figure[15,16].

**Kink from Hund's coupling and its evolution.** An important aspect of the data in Fig. 2 is that a kink feature is observed in the dispersion for all the metallic systems. Such kink feature is normally interpreted to be from electron–phonon coupling. However, the energy scale of the kink, especially for $NiSe_2$, is too large to have a phonon origin. In addition, the kink moves toward the lower energy side as the molar mass becomes lighter (that is, as S content increases), which is opposite to what is expected from the electron–phonon interaction[17–19]. Magnons and plasmons also can be ruled out as the origin of the kink. If the kink originates from electron–magnon interaction, the energy scale of the kink is expected to increase as the system approaches the Mott insulating phase (that is, as S content increases)[20]. On the other hand, plasmons have a much larger energy scale (a few eV or higher) than the kink energy in the data. Another notable aspect of the kink feature is that the kink becomes stronger as the system approaches the Mott insulating phase, alluding to its possible connection to the MIT.

The essential question here is how one can understand the kink feature and its evolution. In order to understand the kink behavior, we first performed DFT+DMFT calculation on $NiS_{1.5}Se_{0.5}$ ($x = 0.5$) compound at 50 K. Figure 3a, c shows the partial Ni $e_g$ orbital spectral function evolution as a function of $U$ without and with $J_H$, respectively. Figure 3b, d indicates the corresponding real part of self-energy $Re\Sigma(\omega)$ of Ni $e_g$ orbital. The behaviors of the spectral function and $Re\Sigma(\omega)$ are found to change quite a lot depending on the presence of $J_H$. Without $J_H$, the $Re\Sigma(\omega)$ follows a quasilinear behavior up to the energy scale of the bandwidth. The slope of $Re\Sigma(\omega)$ increases for a broad energy range as $U$ increases as shown in Fig. 3b. As a result, the overall bandwidth is renormalized and there is no abrupt change in the low-energy scale like in the traditional Brinkman-Rice picture (Fig. 3a).

With the realistic value of $J_H$, however, the overall self-energy behavior is changed as shown in Fig. 3d despite the fact that the effective mass is the same, for example, for ($U = 15$ eV, $J_H =$

0.0 eV) in Fig. 3b and ($U = 5.7$ eV, $J_H = 0.7$ eV) in Fig. 3d. The self-energy follows the Fermi liquid (FL) behavior ($Re\Sigma(\omega) \sim \omega$, $Im\Sigma(\omega) \sim \omega^2$) below a low-energy scale where the abrupt change of $Re\Sigma(\omega)$ results in a kink feature of the spectral function as indicated by the arrows. Beyond the kink energy, $Re\Sigma(\omega)$ shows nearly frequency-independent behavior, so that the overall bandwidth remains almost constant[21] (see also Supplementary Note 7). As $U$ increases, the kink in $Re\Sigma(\omega)$ and spectral function of Ni $e_g$ orbital becomes clear and moves toward the lower energy side. These results indicate that the kink feature found in low-energy scale originates from $J_H$[8,21], and the traditional Brinkman-Rice picture based on one-band Hubbard model should be modified in multiband systems.

Such kink feature has been already discussed in the theoretical description of Hund's metal materials such as ruthenates[7,22–24]. Previous theoretical studies predicted that strongly renormalized coherent part is confined to a low-energy scale set by $J_H$. This explains why the kinks are often found at low-energy scale in Hund's metal[1,7,8,21–25]. This hallmark of $J_H$ survives even in the vicinity of MIT where $U$ is dominant. Our ARPES data in Fig. 2h provide direct spectroscopic information on how the kink induced by $J_H$ evolves as the $U/W$ ratio increases. The kink moves toward the lower energy scale as the system approaches the Mott insulating phase.

We theoretically investigate the evolution of the kink in $NiS_{2-x}Se_x$ using DFT+DMFT calculation for various Se contents. A direct comparison between ARPES and calculated spectral function, however, is difficult due to the small energy scale of the kink and the limitation of analytic continuation. During the analytic continuation, it tends to lose sharp feature in the self-energy which makes it difficult to define the kink in the calculated spectral function. However, one can get important clues from $Re\Sigma(\omega)$ as we discussed above. Figure 3e shows the $Re\Sigma(\omega)$ of Ni $e_g$ orbitals for various Se contents. The narrow region near the Fermi level gets strongly renormalized as the S content increases (that is, as $U/W$ increases due to decreasing W), which is also visible in Fig. 3d with increasing $U$. We define the kink position as the deviation point for $Re\Sigma(\omega)$ from the FL behavior as denoted by arrows in Fig. 3e. In Fig. 3f, we plot the experimentally and theoretically obtained kink positions against the Se content. It is seen that those two results are quite consistent. We also try to define the kink position by using an extremal point of the second derivative of $Re\Sigma(\omega)$, where the band dispersion changes most abruptly. Although the kink positions slightly vary depending on the definition, the overall doping dependent evolution of the kink energy scale does not change (see Supplementary Note 8).

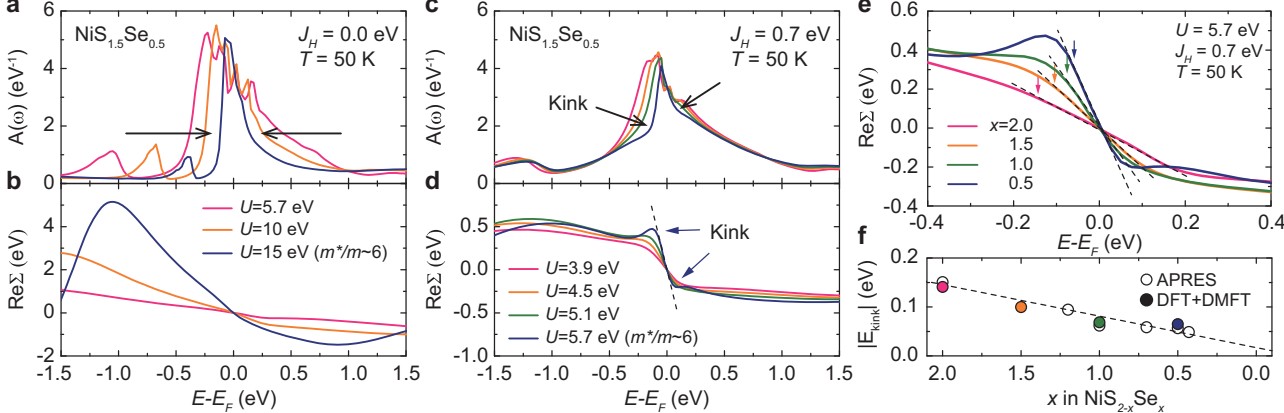

**Fig. 3 Self-energy and spectral function evolution with and without Hund's coupling. a–d** Calculated partial density of states for Ni $e_g$ orbitals and corresponding real part of the self-energy $Re\Sigma(\omega)$ for various $U$ values without and with $J_H$, respectively. **e** Calculated $Re\Sigma(\omega)$ for various Se contents. $Re\Sigma(\omega)$ results are vertically shifted for easier comparison. **f** Kink energy scale obtained from APRES and DMFT self-energy.

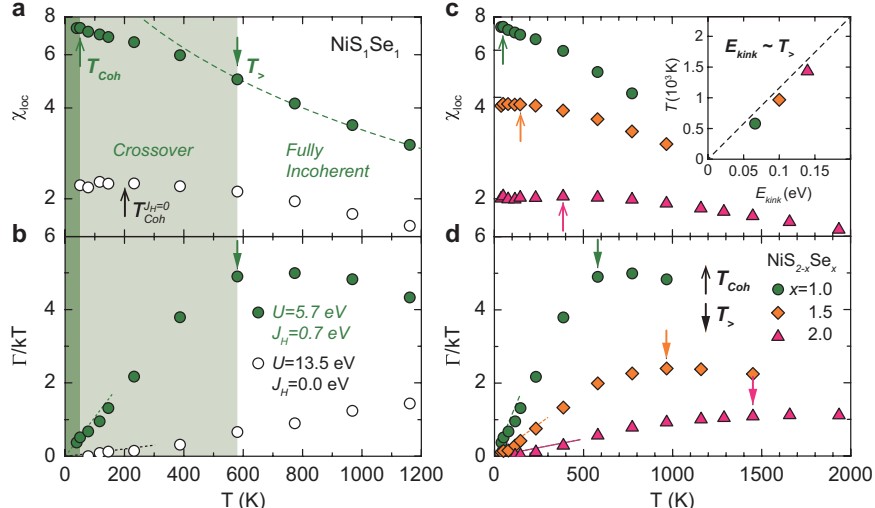

**Fig. 4 Local spin susceptibility $\chi_{loc}$ and inverse quasiparticle lifetime $\Gamma$ as a function of temperature. a, b** Temperature scale analysis of $NiS_1Se_1$ with and without $J_H$, but having the same effective mass. **c, d** Se content-dependent temperature scale analysis. Upward and downward arrows indicate the coherence temperature $T_{Coh}$ and the coherence–incoherence crossover temperature $T_>$, respectively. Below $T_>$, $\chi_{loc}$ deviates from the Curie behavior and $\Gamma/kT$ starts to decrease gradually. Below $T_{Coh}$, $\chi_{loc}$ follows the Pauli behavior and $\Gamma/kT$ shows a linear behavior. The inset indicates the correlation between the kink energy scale obtained from $Re\Sigma(\omega)$, and $T_>$ obtained from $\Gamma/kT$ analysis.

**Kink and characteristic temperature scale.** Now that we have identified the kink structure as a hallmark of $J_H$, we wish to discuss the physical meaning of the kink. Figure 4 shows the local spin susceptibility $\chi_{loc}$ and the inverse quasiparticle lifetime $\Gamma$ as a function of temperature ($\Gamma = -Z\text{Im}\Sigma(i0^+)$). Two temperature scales can be identified from the behavior of $\chi_{loc}$ and $\Gamma/kT$. At high temperature above $T_>$, $\Gamma/kT$ is almost temperature independent, signaling a fully incoherent regime[7]. In this regime, $\chi_{loc}$ follows a Curie behavior ($\chi_{loc} \sim 1/T$), as indicated by the dashed line in Fig. 4a, due to the unscreened local moment. Below $T_>$, $\Gamma/kT$ starts to decreases gradually. Also, the local moment is gradually screened and $\chi_{loc}$ deviates from the Curie behavior. Therefore, it can be defined as the coherence–incoherence crossover temperature[7] or the Kondo temperature[8,25,26]. Below the coherence temperature $T_{Coh}$, the local moment is fully screened and $\chi_{loc}$ follows the Pauli behavior, signaling a FL behavior. Considering the FL behavior of $\Gamma \sim T^2$, $\Gamma/kT$ shows a linear behavior below $T_{Coh}$ (Fig. 4b) at which $\chi_{loc}$ also starts to follow the Pauli behavior (Fig. 4a).

In Fig. 4a, b, we compare the temperature scales of $NiS_1Se_1$ for zero and non-zero $J_H$ while the effective mass is kept the same ($m^*/m \sim 4.2$). For $J_H = 0$, the crossover temperature $T_>^{J=0}$ is found to be located above 1200 K, and $T_{Coh}^{J=0}$ for full recovery of the FL behavior is ~200 K. Meanwhile, the two temperature scales are significantly suppressed ($T_> \sim 600$ K, $T_{Coh} \sim 50$ K) for $J_H = 0.7$ eV. The low coherence temperature observed in the experiment can be explained only by the presence of $J_H$. It is reported that the resistivity of $NiS_{0.67}Se_{1.33}$ ($x = 1.33$), which is less correlated material than $NiS_1Se_1$, follows $T^2$ behavior below ~80 K[15], which is consistent with the suppressed $T_{Coh}$ calculated for $NiS_1Se_1$ in the presence of $J_H$ ($T_{Coh}$ of $NiS_{0.5}Se_{1.5}$ is estimated at ~140 K as shown in Fig. 4c).

In the presence of $J_H$, the $T_>$ is directly related to the deviation from the FL behavior in the self-energy, and its energy scale is in good agreement with the kink position obtained from $Re\Sigma(\omega)$[7,8,25]. Therefore, our observation of the kink in both experiment and theory at low temperature should be directly related to the characteristic temperature scale $T_>$ suppressed by $J_H$[7,8,25,26]. Our system shares some aspects of Hund's physics reported in non-half-filled Hund's metal system. The coherence peak emerges on top of a

broad background rather than inside a pseudo gap as shown in Fig. 3c[8,21,26] and the kink energy scale is directly related to the theoretically extracted temperature scale $T_>$[7,8,25,26]. In addition, the different temperature scales obtained from the calculation with and without $J_H$ indicate that the $J_H$ reduces temperature and energy scales[1,3,4,8,25].

Figure 4c, d shows the calculated $\chi_{loc}$ and $\Gamma/kT$ for various Se contents ($NiS_{1.5}Se_{0.5}$ shows the MIT around 100 K, so it is excluded from the temperature analysis). $T_>$ and $T_{Coh}$ become smaller as the system becomes closer to the MIT with an increasing S content. It is natural that the Kondo resonance and the temperature scale are suppressed as the correlation strength increases. Therefore, the evolution of kink in ARPES can be understood with the evolution of $T_>$ with the correlation strength change. In other words, the suppressed temperature scale by $J_H$ further decreases as $U/W$ ratio increases. Previous optical conductivity measurements showed that the overall kinetic energy of $NiS_{2-x}Se_x$ is suppressed as the system approaches MIT[17]. This is also captured by the theoretical data shown in Supplementary Note 9. The suppression of charge fluctuation with S doping reduces all the energy scales, such as the kink scale, allowing their experimental observation in $NiS_{2-x}Se_x$ system.

In the inset figure, the kink energy scale obtained from $Re\Sigma(\omega)$ at 50 K are plotted against the $T_>$ obtained from the analysis on $\Gamma/kT$. We set the same scale for $x$ (energy) and $y$ ($T$) axes so that the diagonal line indicates a direct conversion between two scales. The graph shows clear correlation between the kink energy scale and the $T_>$, although there could be some error due to the finite temperature points considered in this study. This result demonstrates that the deviation from the FL behavior in $\Sigma(\omega)$ and the kink in spectral function at low temperature are directly projected to the crossover temperature scale $T_>$ of the system.

To summarize, we have directly observed the evolution of the coherence energy scale via kink feature. The ARPES data presented here provide the investigation of how the kink from $J_H$ evolves as the correlation strength increases. From DFT +DMFT calculations, we have confirmed that this kink originates from $J_H$ and is related to the crossover temperature scale $T_>$. The suppression of Kondo screening by $J_H$ makes the kink feature at low-energy scale, and the kink moves toward lower energy side as

the correlation strength further increases by S doping. Our results clearly demonstrate that the evolution of kink can be understood by the evolution of the characteristic energy scale.

## Methods

**Sample information**. Single crystals of NiS$_{2-x}$Se$_x$ ($x = 0.3$, 0.43, 0.5, 0.7, 1.0, 1.2, and 2.0) were obtained by the chemical vapor transport method[27]. The nominal doping levels of the samples are consistent with those estimated from the lattice constants from X-ray diffraction measurements. Their resistivity data are also consistent with the previous results[16,27]: insulating phase for $x = 0.3$, first-order MIT for $x = 0.43$ ($T_{MI} = 24.5$ K) and 0.5 ($T_{MI} = 72$ K), and metallic behavior for $x = 0.7$, 1.0, 1.2, and 2.0.

**Spectroscopy**. High-resolution ARPES spectra by using 60–63 eV photon were obtained at the Beam line 21B1 of the National Synchrotron Radiation Research Center. The energy and angular resolutions were set to 15 meV and 0.1° (corresponds to 0.007 Å$^{-1}$), respectively. The experiments were performed at 13 K and under a vacuum better than $5 \times 10^{-11}$ Torr. ARPES experiments using 100 eV photon were performed at micro ARPES of Beam line 7 of the Advanced Light Source. The energy and angular resolutions were set to 18 meV and 0.1° (corresponds to 0.009 Å$^{-1}$). The 100 eV-ARPES data were taken at 20 K and under a vacuum better than $6 \times 10^{-11}$ Torr. All samples were in situ cleaved along (100) direction under 20 K.

The energy resolution was obtained by fitting the gold spectrum to Fermi–Dirac distribution curve. The inner potential for all doping levels were estimated to be 13 eV in the simple cubic symmetry from photon-energy dependence. FS map was obtained by integrating 15 meV energy range around Fermi level. Figure 1d depicts the momentum integrated spectrum of all the ARPES data that were used to build Fig. 1a.

**Calculation details**. In our calculations, we considered three intermediate compositions $x = 0.5$, 1, 1.5, in addition to the two end-compounds NiS$_2$ ($x = 0.0$) and NiSe$_2$ ($x = 2.0$). A twelve atoms (four Ni atoms and eight chalcogen atoms) unit cell is used for all compositions. For simplicity, we only considered the structures which contain pure chalcogen dimers (i.e., S$_2$ dimers and Se$_2$ dimers, no S-Se dimer). The lattice constants of intermediate compositions were determined by linear interpolations between the experimental lattice constants of two end compounds. For the intermediate compounds, the internal atomic positions are fully relaxed at the DFT level.

To study the correlated electronic structure of NiS$_{2-x}$Se$_x$, we employed a DFT +DMFT calculation as implemented in DFT+ Embedded DMFT (eDMFT) Functional code[28]. DFT calculations were performed by using WIEN2k code which uses the full-potential augmented plane wave method[29]. The Perdew–Burke–Ernzerhof (PBE) generalized gradient approximation (GGA) was used for the exchange-correlation functional[30]. A $12 \times 12 \times 12$ $k$-point mesh was used for self-consistent calculation. The correlation effect of Ni $3d$ orbitals is treated by a DMFT loop on the top of an effective one-electron Hamiltonian generated from the WIEN2k calculation.

The real harmonics basis was used for the local basis for the DMFT calculations. The local axis on Ni atom is also considered to make the hybridization matrix maximally diagonal. The hybridization energy window from $-7$ to 7 eV with respect to the Fermi level was chosen, and $U = 5.7$ eV and $J = 0.7$ eV were used for Ni $d$ orbitals to describe the experimental results. The interaction Hamiltonian is given by the Slater form with the Slater integral $F^0 = U$, $F^2 = (14/1.625)J$, and $F^4 = (8.75/1.625)J$. For the double-counting correction, nominal double-counting method was used[28,31,32]. It has the same form as the fully localized limit method, $E_{dc} = U(n_{cor}^0 - 1/2) - J/2(n_{cor}^0 - 1)$ and $n_{cor}^0$ is the nominal electron occupancy of Ni $3d$ orbitals ($n_{cor}^0 = 8$ in this case). The impurity model was solved by using continuous time quantum Monte Carlo (CTQMC)[33]. To obtain the spectra on the real axis, maximum entropy method is used for analytic continuation.

**Inverse quasiparticle lifetime and local spin susceptibility**. The inverse quasiparticle lifetime $\Gamma$ is given by

$$\Gamma = -Z \text{Im}\Sigma(i0^+) \qquad (1)$$

where $Z^{-1} = 1 - \partial \text{Im}\Sigma(i\omega)/\partial\omega|_{\omega \to 0^+}$. Since we are interested in the energy scale of half-filled $e_g$ orbital, the self-energy of $e_g$ orbitals was used for the $\Gamma$ analysis. The derivative and $\text{Im}\Sigma(i0^+)$ were extracted by fitting a fourth-order polynomial to the data for the lowest ten Matsubara frequencies.

The local spin susceptibility can be defined by

$$\chi_{loc}^{\omega=0} = \int_0^\beta g^2 <S_z(\tau)S_z(0)> d\tau \qquad (2)$$

where $\tau$ is an imaginary time, $g$ is the spin gyromagnetic factor, $S_z$ is the local spin operator, $\beta = 1/(kT)$ is inverse temperature. More details can be found in ref. [33].

## Data availability

The authors declare that the data supporting the findings of this study are available within the paper (and its Supplementary Information).

## Code availability

The DFT+eDMFT code used in this study is available at http://hauleweb.rutgers.edu/downloads/.

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

## Acknowledgements

We are grateful to J.J. Yu, S.R. Park, H.-J. Noh, G. S. Jeon, and Y.K. Bang for fruitful discussions. B.G. Jang would like to thank S.Y. Park for helpful advice. This work was supported by the Institute for Basic Science (IBS) through the Center for Correlated Electron Systems (Grant No. IBS-R009-G2), and also by a National Research Foundation of Korea (NRF) grant funded by the Korea Government (Grant No. 2020R1A5A1019141, No. 2020R1F1A1052898, and No. 2020M3H4A2084418). G.K. was supported by NSF-DMR1733071. CMC was supported by MOST 107-2112-M-213-001-MY3 and NSRRC is operated under The Ministry of Science and Technology, Taiwan. The Advanced Light Source is supported by the Office of Basic Energy Sciences of the U.S. DOE under Contract No. DE-AC02-05CH11231.

## Author contributions

B.G.J., G.H, J.H.S., and C.K. conceived the work. G.H., Y.Y.K., Y.K.K., W.K., and H.D.K. performed ARPES measurements with support from C.M.C. and K.D.T. and analyzed the data. Samples were grown and characterized by G.H., K.D.L., and N.H.; B.G.J., I.P., and D.K. performed the DFT and DFT+DMFT calculations and analyzed the calculation results together with J.H.S. and G.K.; B.G.J., G.H., J.H.S., C.K., and G.K. led the manuscript preparation with contributions from all authors. All the authors discussed the results and commented on the paper.

## Competing interests

The authors declare no competing interests.
