## [Peer Review File · Nature Communications]

REVIEWER COMMENTS

Reviewer #1 (Remarks to the Author):

Jang et al. present in their manuscript “Direct observation of kink evolution with the approach to the metal-insulator transition in $\text{NiS}_{2-x}\text{Se}_x$: Hund’s coupling effect and coherence energy scales” a combined experimental and theoretical study of the effects of Hund’s coupling to the spectral properties of $\text{NiS}_{2-x}\text{Se}_x$ compounds for various S/Se ratios x . From highly resolved ARPES measurements they find clear evidence for kink-like features in the quasi-particle band structure which are tuned by the S/Se ratio and which can be traced until the Mott transition. With the help of accompanying DFT+DMFT calculations they trace the origin of this kink-like feature back to the Hund’s coupling. The manuscript is (mostly) clearly written and seems to present the first decisive findings of J_H induced kink features in Hund’s metals at half filling which clearly deserves publications. However, in its current form I fear that the manuscript does not comply with the high standards of Nature Communications due to the following major and minor points:

The conclusion that the experimentally observed kink-like features are a result of Hund’s coupling is drawn from comparison to DFT+DMFT calculations. The latter are however not described at all. Thus the manuscript lacks (a) reproducibility and (b) clarity. To be able to completely follow the author’s line of argumentation, one needs to understand how the ab initio calculations are performed, how these are linked to the subsequent DMFT calculations, and how the underlying Hamiltonian looks like. It is neither clear what the exact structures are, nor what the “minimal basis” for the correlated subspace looks like, nor how the latter reproduces the DFT results, nor how the double counting problem is treated, nor how the different S/Se ratios are modeled, nor how exactly the DMFT calculations are performed, etc. Due to these missing details, it is impossible to judge whether the remarkable resemblance between the DMFT calculations and experimental results (for example as depicted in Fig. 2K or Fig. 3d) is proof for the realistic level of the DMFT calculation or whether this might be just a lucky coincidence. Questions to answer in this respect are, for example, how the interplay between onsite intra- and inter orbital interactions (U and U'), an orbital-dependent J_h , or a pair hopping J might affect the shown self energies and spectral features? And are onsite, i.e. purely local interactions really enough to describe this system realistically? More over, where do the values of U and J_H come from?

From a comparison to Ref [10] (Moon et al., PRB 92, 235130, 2015) it looks like the presented DMFT calculations from the current manuscript are very similar to the one presented (by some of authors) in the mentioned reference. Again, this is however not clear.

Minor points:

- Phonons are quickly ruled out as a possible origin for these kinks. But how about magnons or plasmons?
- What is the exact effect of the S/Se ratio x ? How is this controlling the Mott transition? Could x have an effect on the interaction matrix elements U , U' , and J_H ? What is the mechanism behind the statement: “By varying the S content, the correlation strength can be easily tuned”?
- Fig. 1a and 1b are hard to compare. It might be beneficial to show them as an overlay.
- What is ξ in Fig. 1c referring to?
- The renormalized DFT band structure in Fig. 1c is claimed to be consistent with the ARPES data. This needs to be explained a bit better. To me it looks like just the Y-M path similar.
- Why is the ARPES data so different in Fig. 1c and 1e? Just due to the different photon energies?

- Where does the phase diagram in Fig. 2a come from? What is the dashed/solid nearly vertical line?

- Fig. 3a,b,c: Which part of the self-energy is shown here? It appears to me that underlying basis has a multi-orbital character. So which orbital channel is shown here?

- What is meant with the statement "In the presence of J_H , the T_c is directly related to the frequency dependence of the self-energy, and its energy scale is in good agreement with the kink position obtained from $\text{Re}S(\omega)$ "?

In conclusion, I need to pronounce that I am in principle in favor for publication of the manuscript by Jang et al. in Nature Communications. The combination of the highly resolved ARPES data together with the presented DMFT calculation / interpretation will be of interest to both experimental and theory communities in the field of strong correlations. However, to this end it must become clear, that the comparison between the experimental data and the DMFT calculations is indeed valid. The latter should become clear by answering the above questions and revising the manuscript accordingly.

Reviewer #2 (Remarks to the Author):

The authors present a combined experimental and theoretical study on the multi band system $\text{NiS}_{2-x}\text{Sex}$ in a wide doping range across the metal-insulator transition. They successfully obtained ARPES spectra in the doping series, which could serve as a model example in which multi-orbital physics can be studied in a systematic manner. In particular, they track the doping evolution of the a band, where they found a kink in the quasiparticle dispersion. Based on the LDA+DMFT calculations, they ascribe the kink to the Hund's coupling.

I would not recommend the publication of this manuscript in Nature Communications due to rather poor agreement between the ARPES data and the DMFT calculations. Subsequently the theoretical claims in the latter half of the manuscript are not substantiated. Note that a more detailed comparison between ARPES and DMFT spectral functions is within the current technology and has already been performed in the ruthenates (see e.g. PRX 9, 021048 (2019) and references therein). There are numerous issues in the manuscript, which I list in the following.

Fig. 1a, b: The agreement between the ARPES intensity and DFT FS is far from consistent. For example, the a band around the G point is almost invisible except for a small portion. The intensity above the letter "Y" is not reproduced by the DFT calculations. The intensity at the right of the "X" looks elliptical. The authors should plot at least the fermi wavevectors to facilitate the comparison. Regarding the FS, Ref. 13 presents a more reliable set of data using bulk-sensitive soft x-ray ARPES.

Fig. 1c: The agreement is rather poor as well. In particular, DFT fails to capture the convex z feature (which they fail to mention in the text), and the energy scale at the X point.

Fig. 1d: The DMFT calculations fail to capture the experimental double peak feature at -1 and -1.5 eV, although their model explicitly includes both t_{2g} and e_g orbitals, casting doubt on their parameter choice. I request the authors to present a direct comparison of ARPES and DMFT spectral functions (in the form of Fig. 1c) and the Fermi surfaces for all the dopings to check the overall consistency.

Fig. 1e and 1f are not mentioned in the text.

Fig. 2: I have a strong doubt on their claim on the presence of "kink" in the correlated phase $x < 1$. There is a strong continuum that overlaps with the quasiparticle dispersion. The "kink" positions they determined in Fig. 1h approximately coincide with the energies where the quasiparticle dispersion and the continuum start to overlap.

Therefore I believe that the apparent bending of peak positions is an artefact originating from the mixture of multiple intensities. Note that the dispersions below the kink for $x=0.7$ and 0.5 are almost vertical, suggesting that they do not reflect the quasiparticle dispersion.

Fig.2k: They present Sommerfeld coefficient from ARPES. How did the authors estimate the contribution from other bands around the M point?

Fig.3c: The link between the energy scales determined here and the experimental kink is incomprehensible. To produce a kink in the spectral function, $\text{Re}S$ needs to have a local maximum as a function of w (as they plot in Fig.3b) and the slight deviation from the w -linear behavior does not produce a kink in the spectral function. Further the lack of k -resolved information (as the authors admit) makes the connection rather weak.

The relevance of the remaining theory part within the combined study cannot be validated unless the above issues have been properly addressed. However, the presented dataset is clearly insufficient to claim "the first direct observation of the evolution of the characteristic temperature scale".

Reviewer #3 (Remarks to the Author):

The authors have identified a suitable and interesting case study for the interplay of U and J_H . From their ARPES spectra they observe a dispersive QP, whose dispersion along k_z supports bulk origins, with a clear kink that is explained as a combined effect of J_H and U - with support from both ARPES and DFT+DMFT. The correlation between the kink energy scale and the cross over temperature $T_>$ further supports the conclusions of the authors.

I found this work quite clear and interesting, and I think it will be well received by the community. The observations are presented in a well ordered way and the data supports the interpretation of J_H being at the origin of the reported QP kink. Kinks have been generally ascribed to correlations or electron-photon interactions so it's quite interesting to see it directly linked to this specific energy scale. I imagine colleagues will be curious to see in which other systems that display QP kinks this interpretation would hold. Hence I think this work meets the requirements for novelty and relevance and I recommend publication in this journal - provided the following corrections are implemented:

figures are missing colorbars and description of some elements (ex: box in panel 1c), the display order of panels in figure 2 is confusing, with b to f then a then h, g, finally i to k. I suggest moving panels a and g to the left.

The photon energy used for ARPES data is not always indicated, alike the temperature at which the spectra were taken. FS should have an indication of which electron energies were integrated/used to make the map. k -integrated spectra should say in which k range.

Generally, symbols used in the figures are not clarified or only clarified in the main text. Would improve readability to correct this.

MDCs acronym is not defined (plus: it's written MDSs, I assume is a typo).

Also, the data used to get panel 2h should be showed (could be a supplementary figure), and the method used to extract the dispersion should be clarified: what function was used? how was the finite resolution taken into account?. I think a sample MDC should be shown, preferably with its fit. All of this could go in the supplementary material.

figure 4: describe Γ evaluation method

The methods should include surface preparation and estimation method for E_F and energy resolution, and for

χ_{loc} .

typos: MDSs, burried

possible broken sentence (or unclear): '... 63 eV photon is utilized for an other $k_z=0$ plane but for the higher resolution.'

figure S2: could indicate the k_z corresponding to 100 and 63 eV photon energy in the figure

figure S3: S5f(g) should be S3f(g)

S4: S7-> S4

Reply to Reviewer #1:

Jang et al. present in their manuscript “Direct observation of kink evolution with the approach to the metal-insulator transition in NiS_{2-x}Se_x: Hund’s coupling effect and coherence energy scales” a combined experimental and theoretical study of the effects of Hund’s coupling to the spectral properties of NiS_{2-x}Se_x compounds for various S/Se ratios x . From highly resolved ARPES measurements they find clear evidence for kink-like features in the quasi-particle band structure which are tuned by the S/Se ratio and which can be traced until the Mott transition. With the help of accompanying DFT+DMFT calculations they trace the origin of this kink-like feature back to the Hund’s coupling. The manuscript is (mostly) clearly written and seems to present the first decisive findings of J_H induced kink features in Hund’s metals at half filling which clearly deserves publications.

We would like to thank the reviewer for a careful and thorough reading of our manuscript and for the constructive suggestions and criticisms that helped us to improve the quality of the manuscript. We are glad to know that the reviewer finds our work significant.

However, in its current form I fear that the manuscript does not comply with the high standards of Nature Communications due to the following major and minor points:

The conclusion that the experimentally observed kink-like features are a result of Hund’s coupling is drawn from comparison to DFT+DMFT calculations. The latter are however not described at all. Thus the manuscript lacks (a) reproducibility and (b) clarity. To be able to completely follow the author’s line of argumentation, one needs to understand how the ab initio calculations are performed, how these are linked to the subsequent DMFT calculations, and how the underlying Hamiltonian looks like. It is neither clear what the exact structures are, nor what the “minimal basis” for the correlated subspace looks like, nor how the latter reproduces the DFT results, nor how the double counting problem is treated, nor how the different S/Se ratios are modeled, nor how exactly the DMFT calculations are performed, etc. Due to these missing details, it is impossible to judge whether the remarkable resemblance between the DMFT calculations and experimental results (for example as depicted in Fig. 2K or Fig. 3d) is proof for the realistic level of the DMFT calculation or whether this might be just a lucky coincidence. Questions to answer in this respect are, for example, how the interplay between onsite intra- and inter orbital interactions (U and U'), an orbital-dependent J_h , or a pair hopping J might affect the shown self energies and spectral features? And are onsite, i.e. purely local interactions really enough to describe this system realistically? More over, where do the values of U and J_H come from?

From a comparison to Ref [10] (Moon et al., PRB 92, 235130, 2015) it looks like the presented DMFT calculations from the current manuscript are very similar to the one presented (by some of authors) in the mentioned reference. Again, this is however not clear.

We thank the reviewer for the detailed comments and suggestions. We understand the concern of the reviewer on the reproducibility and clarity of our calculations. Following the reviewer’s suggestions, we have included more detailed information on the calculation in the ‘Calculation Details’ in the Methods section, and crystal structure information in the Supplementary Materials (S9) so that others can reproduce our calculation result. DFT+DMFT code used in our study is an open-source program and we are also willing to provide our input files if necessary.

For more realistic calculations, we used a real space projector rather than the Wannier method to define the correlated subspace. All valence states within a large energy window (-7 eV to 7 eV) around the Fermi level are kept in the model and are allowed to hybridize with the correlated localized subset. Due to the tilted octahedron, we first defined the local axis on Ni atom so that the hybridization matrix is maximally diagonal. Then the real harmonic basis was used as the local basis for the DMFT calculations (impurity solver) so that the t_{2g} - e_g splitting can be properly described. The entire d -shell were treated dynamically by DMFT part.

Here, we used the density-density form (Ising type) of the Coulomb interaction to reduce the computational cost. *When the same value of interaction parameters are used*, the use of full Coulomb interaction including spin-flip and pair-hopping terms only yields slightly more itinerant electronic structure as shown in figure below (Fig. R1-1). Green and orange lines indicate the self-energy of Ni e_g orbital (at $x=1.0$ case) obtained by using Ising type Coulomb interaction and Full Coulomb interaction, respectively. (Re Σ results are shifted vertically for easier comparison.) The overall behavior of self-energy obtained from two types of Coulomb interaction are almost identical as shown in Fig. R1-1a. As shown in Fig. R1-1b, the use of full Coulomb interaction yields slightly smaller effective mass. In addition, the self-energy obtained from full Coulomb interaction follows the Fermi liquid behavior (Re $\Sigma \propto \omega$, Im $\Sigma \propto \omega^2$) up to higher frequency as indicated by arrows. Therefore, the kink feature occurs at slightly higher energy scale (The energy scales of system become larger). The kink feature can be observed in both type of Coulomb interaction form since the kink originates from the Hund's coupling effect.

Figure R1-1. The self-energy obtained from Full type (orange) and Ising type (green) Coulomb interaction with (a) large energy window (-10 to 10 eV), and (b) narrow region near the Fermi level (-1 to 1 eV).

With the density-density form of Coulomb interaction, we choose the U and J_H values that well describe the phase diagram of $\text{NiS}_{2-x}\text{Se}_x$ (metal-insulator transition as a function of doping x and temperature) [Ref. 10 Moon *et al.*, PRB 92, 235130 2015]. Since the main purpose of our work is to understand and analyze the energy & temperature scales observed in the experiment, it is reasonable to use the U and J_H parameters that can describe the phase diagram consistently. Our choice of parameters also reproduces the experimentally observed effective mass (Fig. 2 k) and the coherence temperature below which the resistivity recovers the Fermi liquid behavior (Fig. 4 b). Therefore, we

believe our choice of parameters is very reasonable. One should use larger interaction parameters with full interaction form of Coulomb interaction to reproduce the experimentally observed phase diagram (MIT) and the coherence temperature scale.

Considering the overall consistency between the experiment and our single-site DMFT calculations in many aspects, local interaction is good enough to describe this system. Besides, energy & temperature analysis based on DMFT calculation was done from $x=2$ to $x=1$ (Fig. 4). In this range, the system shows a paramagnetic behavior for the whole temperature range so that the local interaction is enough to describe the physics of the system. (We excluded $x=0.5$ case in our temperature scale analysis since it shows the MIT at ~ 100 K.)

Minor points:

- Phonons are quickly ruled out as a possible origin for these kinks. But how about magnons or plasmons?

If the kink originates from electron-magnon interaction, the evolution of the kink is expected to be opposite to the experimental observation, much like the phonon case. NiSe_2 is a weakly correlated metal and NiS_2 is a Mott insulator. As explained in the answer to the next question, $\text{NiS}_{2-x}\text{Se}_x$ shows a bandwidth-control Mott transition, and U/W ratio can be tuned by S/Se ratio. Previous neutron diffraction measurement showed that the magnetic moment of $\text{NiS}_{2-x}\text{Se}_x$ decreases as Se content increases [Ref. 27 Matsuura *et al.*, JPSJ 69, 1503 (2000)]. The magnetic moment decreases even more rapidly with Se substitution in the metallic phase (after the metal-insulator transition). This result implies that the magnon energy scale decreases from NiS_2 to NiSe_2 . Previous half-filled Hubbard model study also showed that the magnon energy scale increases as U/W ratio increases [Zheng *et al.*, PRB 72, 033107 (2005)]. Therefore, if the kink originated from magnons, the kink energy scale would increase as the system gets closer to NiS_2 .

Plasmons usually have a much larger energy scale than what is observed in our study, on the order of $\sim eV$. Therefore, the plasmon should be ruled out. We added a short discussion on these two possibilities in the revised manuscript.

- What is the exact effect of the S/Se ratio x ? How is this controlling the Mott transition? Could x have an effect on the interaction matrix elements U , U' , and J_H ? What is the mechanism behind the statement: "By varying the S content, the correlation strength can be easily tuned"?

Figure R1-2 **a**, Molecular diagram of chalcogen dimer. **b-c**, schematic DOS of NiSe₂ and NiS₂, respectively. **d**, DFT+DMFT spectral function of NiS₂

The pyrite system NiS_{2-x}Se_x has long been studied as one of the prototypical half-filled compounds which shows a bandwidth-control Mott transition [M. Imada *et al.*, Rev. Mod. Phys. 70, 1039 (1998)]. In its pyrite structure, two electrons occupy the degenerate Ni *e_g* orbitals due to the strong dimerization of chalcogens, making Ni²⁺ with X₂²⁻ state as shown in the figure above (Fig. R1-2) (**a**, **b**, **d** are taken from Ref. 10). NiSe₂ is a weakly correlated metal with half-filled *e_g* orbitals as shown in Fig. R1-2**b**. On the other hand, the other end compound NiS₂ which is isostructural with NiSe₂ is a Mott insulator. The half-filled *e_g* orbitals split into upper and lower Hubbard bands making a Mott gap (Fig. R1-2**c**). The spectral function of NiS₂ from DFT+DMFT calculation clearly shows the *d-d* gap between the lower and upper Hubbard band (Fig. R1-2**d**).

This is because the *p* orbitals of Se atoms have a larger spatial extent than that of S atoms. Since NiSe₂ has larger *p-d* hybridization and *p-p* transfer interactions, the metallic phase of NiSe₂ can be derived from a Mott insulator by the closing of the Mott gap. The figure below (Fig. R1-3) shows the DFT band structure of NiS₂ and NiSe₂. Red and blue heavy dots indicate the portion of Ni *d* and S/Se *p* orbitals, respectively. For NiSe₂ case, the bandwidth of Ni *e_g* orbitals is larger and the splitting between Ni *e_g* and X₂ *σ** band is smaller (which results in larger *p-d* hybridization). Therefore, we can easily tune the effective correlation strength (*U/W* ratio & *p-d* hybridization) by varying S/Se ratio, *x*.

Figure R1-3 DFT band structure of NiS₂ and NiSe₂. Red and blue heavy dots indicate the portion of

f Ni d and S/Se p orbitals, respectively.

- Fig. 1a and 1b are hard to compare. It might be beneficial to show them as an overlay.

We thank the reviewer for the constructive suggestion. We modified Figure 1a so that readers can compare the ARPES and DFT data more easily. As shown in Fig. 1a (See Figure below – Fig. R1-4), the α band around the Γ point and the ribbon shaped δ bands around the M point well agree with the DFT FS. We have also overlaid the Fermi surface taken with 60 eV photon in the Fig. 1b. The 60 eV data more clearly shows α band around the Γ point.

Figure R1-4 (part of Fig. 1 in main text) Electronic structure of NiSe₂. **a-b**, FS maps from ARPES measurement obtained by using 100 eV and 60 eV photon, respectively. The dashed lines indicate the calculated FS using DFT. **c**, Band structure along the green line in panel **a** from ARPES.

- What is ξ in Fig. 1c referring to?

We thank the reviewer for pointing out our mistake associated with the surface band ζ . We did not include the explanation on ζ band by mistake. As shown in the Fig. 1c, ζ band is reproduced neither by DFT nor by DFT+DMFT calculations. To identify the characteristic of ζ band experimentally, the photon-energy-dependent ARPES was taken, which shows that the ζ band has the surface band character.

We have added a description for the ζ band in the main text and added a new supplementary section. The detailed arguments are shown in Section S2 of the Supplementary Material.

- The renormalized DFT band structure in Fig. 1c is claimed to be consistent with the ARPES data. This needs to be explained a bit better. To me it looks like just the Y-M path similar.

We understand the concern of the reviewer. However, bulk band structure looks rather weak due to the strong intensity of the surface band ζ . Furthermore, the band structure becomes very incoherent below the kink due to the large $\text{Im}\Sigma(\omega)$. Especially, electron-like bulk band near the Γ point becomes almost invisible (see Section S6 in the Supplementary Materials). When we look at Fig. 1a, however, the α and δ bands well agree with the DFT FS. In Fig. 1c, the complicated δ bands give strong intensity near the M point. We have modified the color scale of the map of Fig.1 c so that the bulk originated bands near the M point become more visible. Also, we have added more explanations in the main text.

- Why is the ARPES data so different in Fig. 1c and 1e? Just due to the different photon energies?

As the reviewer correctly understands, the intensity of the ARPES data varies considerably with different photon energies due to the matrix element effect. Indeed, the intensity of $\text{NiS}_{2-x}\text{Se}_x$ ARPES dramatically changes due to the Ni^{2+} fcc structure as discussed in Ref [13]. For such reason, we used two different photon energies to show different aspects of the data: the overall band structure and the α band dispersion. As shown in the figure below (Fig. R1-5), the 106 eV (100 eV for NiSe_2) data is good to show the overall band structure while the α bands are more clearly seen in the 63 eV data (60 eV for NiSe_2). Also, there was a technical benefit to get the wide energy range when we used the 106 eV photon. We have added the Fermi surfaces taken with the two different photon energies in the Supplementary Materials Section S3. For clarity, we have added the NiSe_2 Fermi surface taken with 60 eV in the Fig.1b and more explanation in the main text.

Figure R1-5 (Fig. S3 in supplementary materials) Photon-energy dependent ARPES data along Γ -X direction for $x=0.5$ case. **a**, Fermi surface map for $x=0.5$ in the k_x - k_z plane. **b-c**, Fermi surface maps in the k_x - k_y plane with 106 eV and 63 eV photons, respectively.

- Where does the phase diagram in Fig. 2a come from? What is the dashed/solid nearly vertical line?

The phase diagram of $\text{NiS}_{2-x}\text{Se}_x$ is already well-established. We took the phase diagram from Ref [14]. The phase boundaries for MIT observed in our doping and temperature-dependent resistivity measurement (see Section S1. Sample characterization in the Supplementary Materials) well agree with the previously reported phase diagram of $\text{NiS}_{2-x}\text{Se}_x$. The solid vertical line indicates the transition boundary between AF metal (AFM) and AF Mott insulator (AFI). The dashed line indicates a crossover between paramagnetic metal (PM) and paramagnetic insulator (PI) phase

s, which is not well-defined.

We have added the reference and the explanation for the abbreviation used in the phase diagram in Fig. 2a caption. We also have inserted the new sentence “Our APRES and resistivity (Supplementary Materials S1) results are consistent with the well-known phase diagram of $\text{NiS}_{2-x}\text{Se}_x$ [14]” in the main text.

- Fig. 3a,b,c: Which part of the self-energy is shown here? It appears to me that underlying basis has a multi-orbital character. So which orbital channel is shown here?

Since we focus on the kink feature in Ni e_g bands, the self-energy of Ni e_g orbitals and corresponding partial density of states are shown in Figs. 3 a-c. They are described in the figure caption with the sentence “*Calculated partial DOS of Ni e_g orbitals and corresponding $\text{Re}\Sigma(\omega)$...*”. For clarity, we have added “Ni e_g orbitals” expression throughout the main text.

- What is meant with the statement “In the presence of J_H , the $T_>$ is directly related to the frequency dependence of the self-energy, and its energy scale is in good agreement with the kink position obtained from $\text{ReS}(\omega)$ ”?

We thank the reviewer for pointing out the unclarity associated with the sentence. What we wanted to say is that “the kink energy scale from $\text{Re}\Sigma(\omega)$ (frequency scale) can be translated into the coherence-incoherence crossover temperature scale $T_>$ ”. In the inset of Fig. 4b, we plotted kink energy obtained from $\text{Re}\Sigma(\omega)$ versus the crossover temperature scale $T_>$ obtained from Γ/kT and χ_{loc} analysis. The graph shows the direct conversion between the kink energy scale and $T_>$ temperature scale.

It is now well-established that the Hund’s coupling induces an abrupt change in the self-energy, resulting in the kink feature. The previous theoretical studies on Sr_2RuO_4 suggested that this kink energy scale obtained from the self-energy well agree with the characteristic temperature scale of the system [Ref. 7]. However the previous study only focused on a fixed point so it is difficult to conclude the relation between the kink and the temperature scale. The main purpose of this work is to verify the relation between the kink and the characteristic temperature scale. If the kink is related to the temperature scale of the systems, it should evolve as the correlation strength is varied. For the first time, we observe a kink from Hund’s coupling that evolves as the effective correlation strength varies and is directly related to the temperature scale evolution of the system as shown in Figure 4.

We have modified the sentence as follows.

“In the presence of J_H , the $T_>$ is directly related to the deviation from the FL behavior in the self-energy, and its energy scale is in good agreement with the kink position obtained from $\text{Re}\Sigma(\omega)$ ”

In conclusion, I need to pronounce that I am in principle in favor for publication of the manuscript by Jang et al. in Nature Communications. The combination of the highly resolved ARPES data together with the presented DMFT calculation / interpretation will be of interest to both experimental and theory communities in the field of strong correlations. However, to this end it must become clear, that the comparison between the experimental data and the DMFT calculations is indeed valid. The latter should

become clear by answering the above questions and revising the manuscript accordingly.

We appreciate the reviewer's encouraging assessment that our work will be of interest to both experimental and theory communities in the field of strong correlations. We have improved our manuscript based on the reviewer's comments and suggestions. We believe that our study can deepen the understanding of multiband strongly correlated systems.

Reply to Reviewer #2:

We would like to thank the reviewer for a careful and thorough reading of our manuscript and for the constructive suggestions and criticism that helped us to improve the quality of the manuscript.

The authors present a combined experimental and theoretical study on the multi band system NiS₂-xSex in a wide doping range across the metal-insulator transition. They successfully obtained ARPES spectra in the doping series, which could serve as a model example in which multiorbital physics can be studied in a systematic manner. In particular, they track the doping evolution of the a band, where they found a kink in the quasiparticle dispersion. Based on the LDA+DMFT calculations, they ascribe the kink to the Hund's coupling.

I would not recommend the publication of this manuscript in Nature Communications due to rather poor agreement between the ARPES data and the DMFT calculations. Subsequently the theoretical claims in the latter half of the manuscript are not substantiated. Note that a more detailed comparison between ARPES and DMFT spectral functions is within the current technology and has already been performed in the ruthenates (see e.g. PRX 9, 021048 (2019) and references therein). There are numerous issues in the manuscript, which I list in the following.

Kink features in Hund's metals have already been studied, mostly in ruthenates compounds. However, we respectfully disagree with the reviewer on assessment on the importance of our work. First of all, we would like to point out that no detailed discussion is given on the temperature scale of Hund's metal issue in the PRX paper mentioned by the reviewer. Furthermore, the origin of the kink in ruthenate compounds has been debated. Many previous studies suspected that the kink in the ruthenate compound originates from electron-phonon coupling [Y. Aiura *et al.*, PRL 93, 117005 (2004), H. Iwasawa *et al.*, PRB 72, 104514 (2005), H. F. Yang *et al.*, PRB 93, 121102 (2016)]. After the realization of Hund's physics, there have been several attempts to connect the kink with the Hund's coupling. However, it is difficult to identify the true origin of the kink in the ruthenate compounds.

The critical limitation in previous studies is that they only dealt with the kink at a fixed point (no evolution). It is now well-established that the abrupt change in the self-energy can make the kink feature in spectral functions of multi-orbital systems due to the Hund's coupling. Also, several theoretical studies suggested that this kink can be related to the characteristic temperature scale of the systems. However, there has been no direct evidence for the relation between kink and the characteristic temperature scale. The main purpose of this work is to verify the relation between the kink and the characteristic temperature scale. If the kink is related to the temperature scale of the systems, it should evolve as the

correlation strength is varied. Furthermore, the evolution of the kink can give clear evidence to identify the origin of the kink. This is the starting point of our work, which differentiates our work from previous studies.

However, varying the correlation strength over a wide interaction range in Hund's metal system is found to be difficult. Although the Hund's physics has been discussed mainly in the metallic systems one unit away from half-filling (Hund's metal), the same low-energy effect is expected for the half-filled case in the presence of the Hund's coupling. $\text{NiS}_{2-x}\text{Se}_x$ has long been one of the prototypical half-filled compounds which shows the bandwidth-control Mott transition [M. Imada *et al.*, *Rev. Mod. Phys.* 70, 1039 (1998)]. Therefore we can investigate the Hund's coupling effect over wide interaction range, even up to the Mott metal-insulator transition, which is very difficult in Hund's metal systems.

To the best of our knowledge, this is the first study investigating the Hund's coupling effect in the half-filled multiband system systematically. By varying S/Se ratio of $\text{NiS}_{2-x}\text{Se}_x$, we can investigate the kink evolution depending on the effective correlation strength. The evolution of the kink clearly indicates that the kink observed in our ARPES data is originated from the Hund's coupling rather than electron-boson coupling (e.g., phonon or magnon). In addition, our ARPES and DFT+DMFT results give the direct evidence for the relation between kink and the temperature scale and an important implication that the traditional Brinkman-Rice picture should be modified in multiband systems as we have discussed in our paper.

We understand the reviewer's concern about the comparison between ARPES and DMFT in our work. It does not appear as clear as that in the recent PRX paper. This is because the ruthenate Sr_2RuO_4 is a quasi-2D material and is thus well-suited for ARPES studies while $\text{NiS}_{2-x}\text{Se}_x$ is a 3D material. Furthermore, $\text{NiS}_{2-x}\text{Se}_x$ is close to the Mott metal-insulator transition, unlike Sr_2RuO_4 , and the electronic structure more incoherent. However, please note the following points. (1) The quality of our data is superior to the results of previous PES studies on $\text{NiS}_{2-x}\text{Se}_x$ which enable us to investigate the kink feature for the first time. (2) The quality of our data is good enough to address all the points we are making. The kink energy and Fermi velocity are only two quantities that we are extracting from the experimental data, for which the quality of current data is good enough. (3) $\text{NiS}_{2-x}\text{Se}_x$ has distinct advantages over Sr_2RuO_4 ($\text{NiS}_{2-x}\text{Se}_x$ having doping dependent evolution and being close to a Mott transition) and is different (having e_g orbitals) from Sr_2RuO_4 as described above.

Besides, our DFT+DMFT calculation considered a series of compounds depending on S/Se contents, unlike the recent PRX paper. For the intermediate composition $x=0.5, 1, \text{ and } 1.5$, we used the simplified structure with 12 atoms unit cell. This makes it much more difficult for us to directly compare ARPES data with DFT+DMFT spectral function since the detailed crystal structure affects the electronic structure. However, please note the following points. (1) Without changing U and J_H parameter, our DFT+DMFT calculation well describes the phase diagram of $\text{NiS}_{2-x}\text{Se}_x$ system including the experimentally observed effective mass and the coherence temperature scale. (2) Our DFT+DMFT calculation qualitatively describes the electronic structure and its evolution (See the following responses for details). (3) The self-energy from DFT+DMFT calculation well explains the origin of the kink and its evolution. These cannot be understood in terms of DFT calculation. The main purpose of this work is not to describe all the detailed band structure and the exact kink position in the spectral function, but to explain the origin of the kink and its evolution with DFT+DMFT calculation.

Fig. 1a, b: The agreement between the ARPES intensity and DFT FS is far from consistent. For example, the α band around the Γ point is almost invisible except for a small portion. The intensity above the letter “Y” is not reproduced by the DFT calculations. The intensity at the right of the “X” looks elliptical. The authors should plot at least the fermi wavevectors to facilitate the comparison. Regarding the FS, Ref. 13 presents a more reliable set of data using bulk-sensitive soft x-ray ARPES.

Figure R2-1 (part of Fig. 1 in main text) Electronic structure of NiSe₂. **a-b**, FS maps from ARPES measurement obtained by using 100 eV and 60 eV photon, respectively. The dashed lines indicate the calculated FS using DFT. **c**, Band structure along the green line in panel **a** from ARPES.

We understand the reviewer’s concerns. Some of the ‘inconsistencies’ in Fig. 1 come from the fact that we used 100 eV photon for the map. Due to photoemission cross section, the α band is weak in the first Brillouin zone (near the center of Fig. 1a). In fact, α band is more clearly seen in the second Brillouin zone (upper-left and upper-corner of the figure). On the other hand, the α band is more clearly seen in the 60-63 eV data in Supplementary Fig. 3c and agrees well with the data in Ref. 13 as well as our DFT FS. Therefore, we used 60-63 eV data to analyze the kink feature in the α band. (We have also overlaid 60 eV Fermi surface map in Fig. 1 b for the better comparison with 100 eV data.)

The reason we used 100 eV data in Fig. 1 is to have a wide energy and momentum range (for a technical reason) for a comparison of overall features. For a better comparison between Figs. 1a and 1b, we have overlaid the DFT FS on APRES FS in Fig. 1a as the reviewer suggested. The α band around the Γ point and the ribbon shaped δ bands around the M point well agree with ARPES FS. Especially, the calculated α band FS matches the experimental data well in the second Brillouin zone and also the measured α band in Fig. S3. The ARPES data set presented in Ref. 13 does not show the detailed shape of δ bands around M point.

Figure R2-2 Photon energy dependent APRES data of NiSe₂

As for the extra intensity near X and Y points (black arrow in Fig. R2-2a) which is not accounted for by the bulk calculation, it is from surface bands (arrows in Fig R2-2c. Also, refer to the electron pockets seen in the second BZ as described in the last paragraph of S2). To identify the characteristic of the band, photon-energy-dependent ARPES was performed. As indicated by the white arrow in R2-2b, the elliptical band near the X point has no photon energy dependence. Even though the intensity varies due to the matrix element effect, the band structure is independent of the incident photon energy. The intensity above the Y point is from the same band but the matrix element is different due to the difference in the incidence angle. This assignment is consistent with the bulk-sensitive soft X-ray ARPES work in Ref. 13, in which the bands near X and Y points were not seen. The detailed arguments for the surface origin of the band are given in Supplementary Material S2.

Fig. 1c: The agreement is rather poor as well. In particular, DFT fails to capture the convex z feature (which they fail to mention in the text), and the energy scale at the X point.

We thank the reviewer for pointing out our mistake associated with the surface band ζ . We did not include the explanation about the ζ band by mistake. As shown in Fig. 1c, ζ band is reproduced neither by DFT nor by DFT+DMFT calculations. To identify the characteristic of ζ band experimentally, the photon-energy-dependent ARPES was taken and the result shows that ζ band has the surface band character. The detailed arguments are given in S2 of the Supplementary Material.

With photon energy of 100 eV, the surface band ζ is relatively strong and thus tends to make the bulk bands obscure. For example, the electron-like bulk band near the Γ point at ~ 0.3 eV is not almost invisible due to such effect in addition to the large $\text{Im}\Sigma(\omega)$ which is the characteristic feature of Hund's metal system (See Fig. S6 and the references in the main text). Yet, the α and δ bands around the M point are found to be consistent with the DFT FS. We also have modified the color scale of the map of Fig.1c so that the bulk bands near the M point become more visible and can be compared with DFT bands. We have added more explanation in the main text.

Fig. 1d: The DMFT calculations fail to capture the experimental double peak feature at -1 and -1.5 eV, although their model explicitly includes both t_{2g} and e_g orbitals, casting doubt on their parameter choice.

As the reviewer pointed out, the experimental spectral function shows double-peak feature at around 1.1 and 1.4 eV binding energies. We have already noticed this double peak feature in t_{2g} band. This splitting comes from the slightly distorted octahedral environment of Ni- Se_6 . Due to the distortion of the octahedron, the t_{2g} bands further split into doubly degenerate e_g^π states and a single a_{1g} state (See figure below – Fig. R2-3a). The e_g states, which are of our interest, are not affected by this distortion.

In our DFT+DMFT calculation, the crystal electric field energy levels are first calculated on the level of the lattice (i.e. in the DFT part) and then they are renormalized inside the impurity solver (by the DMFT self-energy). In our original calculation, we have used t_{2g} - e_g basis (real harmonic basis) as a local basis for the DMFT calculation due to the small distortion. As a result, the whole t_{2g} levels (a_{1g} and e_g^π) are treated equally in the impurity solver, although a_{1g} and e_g^π splitting is treated in the DFT part with first principle manner.

We have already checked the calculation with a more precise basis (diagonal basis), which can describe a_{1g} and e_g^π states separately inside the impurity solver (DMFT part), as shown in the following figure (Fig. R2-3b). The total spectral functions from two bases are almost identical, indicating the validity of t_{2g} - e_g basis (real harmonic basis). Although the double peak feature is not well captured in the total density of states (DOS) of DFT+DMFT calculation, the orbital resolved DOS clearly shows the splitting between a_{1g} and e_g^π states, which are located at -1.15 eV and -1.4 eV. This calculated splitting agrees well with the experimentally observed double-peak structure. Please note that e_g states which are of our interest are not affected by the basis choice. Our parameter choice describes not only the phase diagram of NiS_{2-x}Se_x well but also the experimentally observed effective mass (Fig. 2 k) and the coherence temperature below which the resistivity recovers a Fermi liquid behavior (Fig. 4b and related main text). Therefore, we believe our parameter choice is valid.

Figure R2-3 a, Crystal field splitting of d orbital depending on the local symmetry **b**, Calculated DFT+DMFT DOS with two different basis.

I request the authors to present a direct comparison of ARPES and DMFT spectral functions (in the form of Fig. 1c) and the Fermi surfaces for all the dopings to check the overall consistency.

Figure R2-4 a-b, FS map for $x=2.0$, 0.7 , and 0.5 obtained by using 100 (106) eV and 60 (63) eV photon, respectively. **c-e**, k-resolved spectral function from ARPES (left panels) and DFT+DMFT (right panels).

We have 3 doping cases ($x=2.0$, 0.7 , 0.5) for the full spectral function in the form of Fig. 1c with photon energies in the 100-106 eV range. DFT+DMFT calculation was conducted at every $x=0.5$ points, so that the only two cases ($x=2$, 0.5) can be directly compared with the ARPES data as shown in the above figure. For $x=0.7$ case, we overlay the renormalized DFT band structure of $x=1.0$ case, and the renormalized factor is obtained from the interpolation between $x=0.5$ and 1.0 case.

Figure R2-4 **a** and **b** show the Fermi surface (FS) map obtained with 100 (103) eV and 60 (63) eV

photon energy, respectively. As we discussed above, overall FS are well captured by 100-103 eV data while the α band is more clearly seen in the 60-63 eV data. The dotted lines represent the calculated FS. The rounded hexagon shape of α band FS at $x=2.0$ and $x=0.5$ case are well reproduced by the calculated FS.

Figure R2-4 **c-e** show the band structure obtained from ARPES measurement (left panels) and the corresponding DFT+DMFT result (right panels). At $x=2.0$ case (figure **c**), as we discussed above, the complex δ bands around M point and the α band near Y point are well described by DFT+DMFT calculation. The band dispersion along Y(X)-M direction also well agrees with each other. At $x=0.5$ case (figure **e**), DFT+DMFT calculation well describes the strongly renormalized band structure near the Fermi level. The detailed band structure cannot be clearly resolved since $x=0.5$ case is close to the MIT. However, the band dispersion along Y(X)-M direction well agrees with each other.

Please note that the coherent part is confined within a low energy range near the Fermi level due to the Hund's coupling effect (See Fig. S6 and the references in the main text). At $x=2.0$ case, the DMFT spectrum becomes very incoherent beyond ~ 0.2 eV. This genuine coherent part near the Fermi level becomes much narrower in $x=0.5$ case since the energy window where the self-energy follows the Fermi liquid behavior becomes narrower (kink moves toward to lower energy side). Although the e_g band actually extends to about -0.5 eV, the spectral function between -0.5 and -0.1 eV becomes very incoherent due to the large $\text{Im}\Sigma(\omega)$ which comes from the Hund's coupling effect (See Supplementary S6). This behavior is also well captured in ARPES data. The genuine coherent regime becomes narrower as the x decreases. Although the renormalized DFT band structure (dotted lines) are well agree with ARPES and DFT+DMFT result at this coherent regime near the Fermi level, but it deviates from the DFT+DMFT result above and below this coherent regime because the renormalization factor (the slope of $\text{Re}\Sigma$) changes at the kink (deviation from the Fermi liquid behavior). This deviation (kink) is more clearly seen above the Fermi level, especially at $x=0.5$ case (figure **e**). As we mentioned in our main text, the kink becomes clear and moves towards the lower energy side as the system become closer the MIT (as S content increases).

In the $x=2.0$ case (NiSe_2), the additional electron like pocket around the Γ point is observed in the DFT+DMFT result. This very coherent band has mainly Se p character as shown in the figure below. Figure R2-5 shows the DFT band structure of NiSe_2 . Red and blue heavy dots indicate the portion of Ni d and Se p orbitals, respectively. In DFT calculation, The conduction band minimum at Γ point has mainly Se p orbital character. The interaction (bonding) between Se and Ni atoms is somewhat overestimated at the Γ point in the DFT+DMFT calculation so that the band moves below the Fermi level. Although this Se p band is not observed in the ARPES data, the Ni e_g bands, which are of our interest, are well described by DFT+DMFT calculation. Since we are interested in the energy scale of the system, we focused on describing the effective mass of Ni d orbitals and the metal-insulator transition observed in the experiment.

This overestimation of the interaction (bonding) between Se and Ni atoms is already reported in a previous theoretical study [J. Kunes *et al.*, PRB 81, 035122 (2010)]. In the paper, the authors show that the DFT overestimates the Se_2 dimer bond length even though it predicts the S_2 dimer bond length. The overestimation of Ni-Se interaction (bonding) gives shorter Ni-Se bonding, which results in a longer Se_2 dimer bond length due to the crystal symmetry. One may need more sophisticated method like GW+DMFT for the exact description of Se p band in this system. However, the overall ARPES data are qualitatively well described by our DFT+DMFT calculation.

Figure R2-5 DFT band structure of NiSe₂. Red and blue heavy dots indicate the portion of Ni *d* and S/Se *p* orbitals, respectively. The conduction band minimum at Γ point has mainly Se *p* orbital character.

Fig. 1e and 1f are not mentioned in the text.

Fig. 1e and 1f were already described in the text with the sentence “Figures 1e and 1f show the α band of NiSe₂ obtained from ARPES and DFT+DMFT, respectively”.

We have modified the sentence as follows.

“Figures 1e and 1f show the ARPES and DFT+DMFT results, respectively, of the NiSe₂ alpha band from the red boxed region in Fig. 1c.”

Fig. 2: I have a strong doubt on their claim on the presence of “kink” in the correlated phase $x < 1$. There is a strong continuum that overlaps with the quasiparticle dispersion. The “kink” positions they determined in Fig. h approximately coincide with the energies where the quasiparticle dispersion and the continuum start to overlap. Therefore I believe that the apparent bending of peak positions is an artefact originating from the mixture of multiple intensities. Note that the dispersions below the kink for $x = 0.7$ and 0.5 are almost vertical, suggesting that they do not reflect the quasiparticle dispersion.

As we discussed above, there is a surface band ζ along the Γ -M direction right below the α band. Although there is some overlap between the α and broad ζ bands, our ARPES data clearly distinguish them. In Fig. 2g, quasiparticle peaks are clearly distinguished from the broad continuum that comes from the surface band ζ and the incoherent spectral weight due to the electron-electron correlation. We understand the reviewer’s concern so that we have included a detailed description on how we obtained the quasiparticle dispersion using the momentum distribution curve (MDC) fitting method in the Supplementary Materials. In Fig.S5, the α band is clearly distinguished from the surface band ζ up to a high energy scale than the kink energy scale.

Furthermore, one of the main features of Hund’s metal system is that the genuine coherent part, but strongly renormalized part, is confined to the low-frequency scale (below the kink), while the band dispersion becomes very incoherent at high energy scale (See Fig. S6 and the references in the main

text). Therefore, the band dispersion above the kink energy scale becomes not as clear as the dispersion near the Fermi level. The work on Sr_2RuO_4 in the recent PRX paper also shows the same behavior. They defined the kink energy scale as 20 meV where the real part of the self-energy starts to deviate from the linear behavior. Although their ARPES data shows the band dispersion near the Fermi level very clearly as the reviewer pointed out, the band dispersion becomes very incoherent above the kink energy scale, and it is not well defined above the 30~40 meV as seen in Fig.3f of the PRX paper. The elusive band dispersion at a high frequency above the kink feature is the result of Hund's coupling effect. Therefore, it is not a coincidence that the kink feature is located near the incoherent regime ("the continuum" mentioned by the reviewer) but a result of the Hund's coupling effect, even though there is also some overlap with the surface band.

Fig.2k: They present Sommerfeld coefficient from ARPES. How did the authors estimate the contribution from other bands around the M point?

The Sommerfeld coefficient γ shown in Fig. 2 k was taken from the previous study [Ref. 14], not obtained from our ARPES data. Since the bands around the M point are complex and the bottom of the bands lie just below the Fermi level, it is difficult to extract γ from the experimental δ bands. Despite the fact that the δ contribution was not included in the discussion, we believe that the α band contribution is sufficient to discuss overall transport properties in this system as mentioned in the main article.

In Fig. 2k, we want to show that the effective mass of the system undergoes the first-order transition across the MIT transition, which is not consistent with the conventional bandwidth controlled MIT in the single-band system (therefore, whether it diverges or not near MIT is what is important, not the exact value). The blue line in Fig. 2k indicates the effective mass obtained by comparing the α band dispersion between ARPES and DFT results. The red line shows the effective mass obtained from the DMFT electron self-energy. The effective mass obtained from two different methods well agree with each other and they do not diverge at the MIT transition. To verify the finite effective mass at the MIT, we compare the effective mass evolution with the behavior of the Sommerfeld coefficient from the previous study which also does not show a divergent behavior at the MIT.

Fig.3c: The link between the energy scales determined here and the experimental kink is incomprehensible. To produce a kink in the spectral function, $\text{Re}\Sigma$ needs to have a local maximum as a function of w (as they plot in Fig.3b) and the slight deviation from the w -linear behavior does not produce a kink in the spectral function. Further the lack of k -resolved information (as the authors admit) makes the connection rather weak.

We thank the reviewer for bringing up this point. We understand that the band dispersion changes most abruptly where $\text{Re}\Sigma$ has a local maximum. However, $\text{Re}\Sigma$ is the 'difference' between the measured and unrenormalized dispersions. Therefore, contrary to what the review stated, one does not need a local maximum $\text{Re}\Sigma$ to have a kink. The band dispersion "starts" to bend (that is, kink) when $\text{Re}\Sigma$ starts to deviate from the linear behavior (renormalization factor starts to change). The previous theoretical study, which deals with the kink and the energy scale, also defines the kink position with the deviation from the Fermi liquid behavior (linear behavior in $\text{Re}\Sigma$) [Ref. 7, 22, 23, 24]. From the point of view of the physical meaning, the deviation from the Fermi liquid behavior in $\text{Re}\Sigma$ is directly related to the temperature scale of the system as we discussed in our manuscript. Because of these reasons, we define the kink position with the deviation point from the linear behavior rather than the local maximum

point in $\text{Re}\Sigma$.

Following the reviewer's suggestion, we tried to define our kink position at which $\text{Re}\Sigma$ changes most abruptly by using the second derivative of $\text{Re}\Sigma$ (Due to the numerical problem, the smoothing is included when the second derivative of $\text{Re}\Sigma$ is calculated). The vertical dashed lines in Fig. R2-6 **b** indicate the kink defined from the second derivative of $\text{Re}\Sigma$ ($\text{Re}\Sigma''$). In this way, the kink energy scale obtained from DFT+DMFT calculation is slightly higher than that of ARPES (The colored horizontal bidirectional arrows indicate the deviation between the kink defined from the deviation and $\text{Re}\Sigma''$). However, they are just vertically shifted up from the kinks defined from the deviation from the linear behavior of $\text{Re}\Sigma$ (or the ARPES results) as shown in Fig. R2-6 **c**. Overall doping dependent evolution of the energy scale does not change (the slope remains almost the same). Regardless of the way how to define the kink position, our DFT+DMFT calculation well describes the evolution of the energy scale as a function of the doping content x .

Figure R2-6 a-b, Real part of self-energy $\text{Re}\Sigma$ and its second derivative $\text{Re}\Sigma''$. The vertical lines indicate the kink position defined with $\text{Re}\Sigma''$. **c**, kink energy scale obtained from ARPES, $\text{Re}\Sigma$ (Deviation from Fermi liquid behavior), and $\text{Re}\Sigma''$ (local maximum).

The relevance of the remaining theory part within the combined study cannot be validated unless the above issues have been properly addressed. However, the presented dataset is clearly insufficient to claim “the first direct observation of the evolution of the characteristic temperature scale”.

We respectfully disagree with the reviewer on this conclusion and we believe we provided evidence and explanations to prove our points.

Reply to Reviewer #3:

The authors have identified a suitable and interesting case study for the interplay of U and J_H . From their ARPES spectra they observe a dispersive QP, whose dispersion along k_z supports bulk origins, with a clear kink that is explained as a combined effect of J_H and U - with support from both ARPES and DFT+DMFT. The correlation between the kink energy scale and the cross over temperature T_c further supports the conclusions of the authors.

I found this work quite clear and interesting, and I think it will be well received by the community. The observations are presented in a well ordered way and the data supports the interpretation of J_H being at the origin of the reported QP kink. Kinks have been generally ascribed to correlations or electron-photon interactions so it's quite interesting to see it directly linked to this specific energy scale. I imagine colleagues will be curious to see in which other systems that display QP kinks this interpretation would hold. Hence I think this work meets the requirements for novelty and relevance and I recommend publication in this journal - provided the following corrections are implemented:

We would like to thank the reviewer for a careful and thorough reading of our manuscript and for the constructive suggestions and criticism that helped us to improve the quality of the manuscript. We very much appreciate the reviewer's recognition of the novelty and importance of our work and encouraging assessment that our paper is: clear and interesting, will be well received by the community. We tried to improve our manuscript based on the reviewer's comments and suggestions.

figures are missing colorbars and description of some elements (ex: box in panel 1c), the display order of panels in figure 2 is confusing, with b to f then a then h, g, finally i to k. I suggest moving panels a and g to the left.

We thank the reviewer for the constructive suggestion. Following the reviewer's suggestion, we have revised Fig. 2 in the new manuscript.

The photon energy used for ARPES data is not always indicated, alike the temperature at which the spectra were taken. FS should have an indication of which electron energies were integrated/used to make the map. k -integrated spectra should say in which k range.

Following the reviewer's suggestion, we have added the photon energy and temperature in the figure and main text. Also, we have added a short description in the method section as follows:

FS map was obtained by integrating 15 meV energy range around the Fermi level. k -integrated spectra of Fig. 1d is obtained by summing all the energy distribution curves of ARPES data in Fig. 1a.

Generally, symbols used in the figures are not clarified or only clarified in the main text. Would improve readability to correct this.

We tried to improve the readability by clarifying the symbols used in the figures.

MDCs acronym is not defined (plus: it's written MDSs, I assume is a typo).

Also, the data used to get panel 2h should be showed (could be a supplementary figure), and the method used to extract the dispersion should be clarified: what function was used? how was the finite resolution taken into account?. I think a sample MDC should be shown, preferably with its fit. All of this could go in the supplementary material.

Following the reviewer's suggestion, we have added a new supplementary section describing how we obtained the quasiparticle dispersion from ARPES data. The quasiparticle α band is characterized by the Lorentzian while the surface band ζ is characterized by a Gaussian peak. The α band is well distinguished from the surface band and incoherent background up to a higher energy scale than the kink energy scale. More details are shown in Section S5 of the updated Supplementary Material.

figure 4: describe \Gamma evaluation method

We have introduced a short description in the main text, " $\Gamma = -Z\text{Im}\Sigma(i0^+)$ ", where $Z = (m^*/m)^{-1}$ ". We also explained the detailed evaluation method for Γ and χ_{loc} in the 'Methods' section.

The methods should include surface preparation and estimation method for E_F and energy resolution, and for χ_{loc} .

We have updated the method section in the revised manuscript. All samples were in situ cleaved along (100) direction under 20 K. The photon energy and the energy resolution are observed by fitting the gold spectrum to the Fermi-Dirac distribution curve.

possible broken sentence (or unclear): '... 63 eV photon is utilized for an other $k_z=0$ plane but for the higher resolution.'

We have corrected the sentence as '... 60 eV photon is utilized for clearer observation of the α band with the higher resolution.' (We have noticed our mistake in the photon energy description. 60 eV photon was used to measure NiSe₂ sample while 63 eV photon was used for other doping levels.)

figure S2: could indicate the k_z corresponding to 100 and 63 eV photon energy in the figure

Following the reviewer's suggestion, we have updated the figure in the revised manuscript.

figure S3: S5f(g) should be S3f(g)

S4: S7-> S4

It is fixed.

REVIEWER COMMENTS

Reviewer #1 (Remarks to the Author):

The authors adequately answered and responded to most of my questions and remarks. However, they missed the chance to give clear answers / responses to the following previously raised points:

1.) From a comparison to Ref [10] (Moon et al., PRB 92, 235130, 2015) it looks like the presented DMFT calculations from the current manuscript are very similar to the one presented (by some of authors) in the mentioned reference.

2.) Could x have an effect on the interaction matrix elements U , U' , and J_H ?

Though I would have appreciated clear statements on these two points, I anyway understood from the whole reply to all referees including the additionally supplied data, that

(a) the model parameters are adequate to reproduce the MIT at $x=0.5$ as well as to reasonably reproduce the spectral function for NiSe₂, i.e. $x=2.0$. Thus, one might argue that they describe the whole x -dependent phase space reasonably well.

(b) the main finding of the manuscript at hand is to verify the distinct relation between the temperature scale T_K and the kink positions, which in turn is induced by J_H and controlled by the continuously tuned correlation strength via x .

The latter is shown in the inset of Fig. 4 b which correlates the kink position and T_K , which have both been analyzed in the manuscript as a function of x . The kink positions has also been experimentally verified. These are indeed highly relevant findings which, in principle, deserve to my mind publication in Nature Communications. However, based on the response by the authors to all referee remarks I have a few further questions / remarks which should be answered before the manuscript can be published:

1.) The data presented in Fig. 3 d plays an essential role since it is a major ingredient to the main findings presented in the inset of Fig. 4 b (fully theoretical) and underlines the validity of the theoretical results by comparison to the experimental values. At a first glance the agreement of the experimental and theoretical kink positions is remarkable. However, from the supplied additional data it seems to me that the theoretical kink position is also dependent on the model parameters (full vs. Ising Coulomb interactions, as shown in Fig. R1-1) as well as on the definition of the kink position (see Fig. R2-6 and discussion around it). To me it is perfectly fine that this agreement is indeed not perfect (there is no full predictive power of DFT+DMFT calculations yet), but this should be clearly acknowledged, preferably inform of error bars and / or a small discussion about possible origins of deviations.

2.) The second important (theoretical) ingredient to the data presented in the inset of Fig. 4 b is T_K as a function of x as shown in Fig. 4 b. It is, however, not very clear how T_K is extracted and how reliable this extraction is. I.e., shouldn't there be additional error bars for T_K ? Furthermore, it would be beneficial to know if there is any experimental data in line with these numbers.

3.) Based on Fig. R1-1 and R2-3 (b) I could imagine that the (theoretical) effective masses shown in Fig. 2 k should also carry some error bars (the eg channel in R2-3 (b) changes slightly around EF with the change in the basis set). The authors should comment on this.

Reviewer #2 (Remarks to the Author):

The authors carefully addressed my questions and suggestions in the revised manuscript. I additionally suggest that they include their response Figure R2-4 in the supplementary materials for the reader's convenience. After this minor correction I would recommend the publication of this manuscript.

Reviewer #3 (Remarks to the Author):

I confirm my recommendation for this work to be published in this journal. The authors did a good job and the changes implemented meet my requests, however I did find a few things that were not fully addressed:

1) Fig 1: The relevant compound (NiSe₂) could be mentioned in the figure

2) clarification: the k-integrated spectra for panel 1d are those shown in panel 1c or 1a? the text says 1a in more than one place, but 1a is a 2D fermi surface map, I suspect it might be a typo? if not, could you please further illustrate the integration method?

3) Fig. 2 caption: buried is misspelled (buried)

4) figure 2 panel h: are the experimental data -from whose analysis were extracted the reported results - shown in the article or SI?

The caption only reads: "QP dispersions obtained by fitting the momentum distribution curves (MDCs)." without stating from which spectra these MDCs were taken.

Here section S5 could be referred to, for further info on the fitting, with the addition of a clarification as to what cuts were used, and if those are shown in the text.

I am guessing the analyzed spectra were 2b-e, while dopings 1.2 and 1.0 are not shown. All spectra analyzed but not shown in the main text should be reported at least in the SI.

5) figure 4: in panel a is indicated the compound (NiS₁Se₁), it would be good to do the same for panel b, since different dopings are shown. It could go near the legend.

Of these, only point 2 and 4 are critical, while the others are more aesthetic comments.

Reply to Reviewer #1:

We greatly appreciate the detailed comments by the reviewer. We also would like to thank the reviewer for recognizing the importance of our work.

The authors adequately answered and responded to most of my questions and remarks. However, they missed the chance to give clear answers / responses to the following previously raised points:

1.) From a comparison to Ref [10] (Moon et al., PRB 92, 235130, 2015) it looks like the presented DMFT calculations from the current manuscript are very similar to the one presented (by some of authors) in the mentioned reference.

We used basically the same calculation scheme as the one in Ref [10], although we recalculated all the calculations shown in this study (since we used later versions of DFT and DMFT programs in this study, there can be some discrepancy between the results of this study and Ref [10].) In Ref [10], we had successfully described the phase diagram of $\text{NiS}_{2-x}\text{Se}_x$ and studied the origin of the metal-insulator transition. Here, we focused on the origin of the kink feature and the energy scale of $\text{NiS}_{2-x}\text{Se}_x$ based on the parameters verified in Ref [10].

2.) Could x have an effect on the interaction matrix elements U , U' , and J_{H} ?

Though I would have appreciated clear statements on these two points, I anyway understood from the whole reply to all referees including the additionally supplied data, that

(a) the model parameters are adequate to reproduce the MIT at $x=0.5$ as well as to reasonably reproduce the spectral function for NiSe_2 , i.e. $x=2.0$. Thus, one might argue that they describe the whole x -dependent phase space reasonably well.

Although we have not systematically studied the effect of doping x on the interaction parameters (using, for example, constrained random phase approximation cRPA), the interaction parameters might slightly vary depending on doping x . However, we used the same interaction parameters for all x values to avoid the ambiguity of parameter tuning. As the reviewer pointed out, the interaction parameters used in our study well describe not only the phase diagram of $\text{NiS}_{2-x}\text{Se}_x$ but also the experimentally observed effective mass (Fig. 2 k) and the coherence temperature below which the resistivity recovers a Fermi liquid behavior (Fig. 4b and related main text). Therefore, we believe our parameter choice is valid for the analysis of the energy scale of $\text{NiS}_{2-x}\text{Se}_x$.

(b) the main finding of the manuscript at hand is to verify the distinct relation between the temperature scale T^* and the kink positions, which in turn is induced by J_{H} and controlled by the continuously tuned correlation strength via x .

The latter is shown in the inset of Fig. 4 b which correlates the kink position and T^* , which have both been analyzed in the manuscript as a function of x . The kink positions has also been experimentally verified. These are indeed highly relevant findings which, in principle, deserve to my mind publication in Nature Communications. However, based on the response by the authors to all referee remarks I have a few further questions / remarks which should be answered before the manuscript can be published:

We are glad to know that the reviewer recognizes the importance of our work.

1.) The data presented in Fig. 3 d plays an essential role since it is a major ingredient to the main findings presented in the inset of Fig. 4 b (fully theoretical) and underlines the validity of the theoretical results by comparison to the experimental values. At a first glance the agreement of the experimental and theoretical kink positions is remarkable. However, from the supplied additional data it seems to me that the theoretical kink position is also dependent on the model parameters (full vs. Ising Coulomb interactions, as shown in Fig. R1-1) as well as on the definition of the kink position (see Fig. R2-6 and discussion around it). To me it is perfectly fine that this agreement is indeed not perfect (there is no full predictive power of DFT+DMFT calculations yet), but this should be clearly acknowledged, preferably inform of error bars and / or a small discussion about possible origins of deviations.

We thank the reviewer for the constructive suggestion. Following the reviewer's suggestion, we have added a short discussion about the possible origin of deviations in the main text and the supplementary materials (Figure S8-2 and related discussion).

2.) The second important (theoretical) ingredient to the data presented in the inset of Fig. 4 b is $T^>$ as a function of x as shown in Fig. 4 b. It is, however, not very clear how $T^>$ is extracted and how reliable this extraction is. I.e., shouldn't there be additional error bars for $T^>$? (1) Furthermore, it would be beneficial to know if there is any experimental data in line with these numbers. (2)

(1) As mentioned in the main text and calculation details part, $T^>$ is defined from the inverse quasiparticle lifetime Γ ($\Gamma = -2\text{Im}\Sigma$). The derivative and $\text{Im}\Sigma(i0^+)$ were extracted by fitting a fourth-order polynomial to the data for the lowest ten Matsubara frequencies. From the Γ/kT analysis, $T^>$ is defined as the temperature at which Γ/kT becomes temperature independent, which is a signal of a fully incoherent regime. Although there could be some error for the estimated Γ from the Monte Carlo sampling noise, this error is small (See Fig. R1-1a below), so that $T^>$ is not changed with the temperature points considered in this study. There can be additional error since we cannot calculate the Γ for the whole temperature points (we can only sample finite temperature points). Therefore, there could be some error in the temperature axis. Since it is not rigorous to define the error bar here, we have included a brief comment on the possible error as follow.

"The graphs show a clear correlation between the kink energy scale and the $T^>$, although there might be some error due to the fact that finite number of temperature points were considered in this study."

(2) Unfortunately, there is no experimental data to support the theoretically obtained $T^>$ so far. From the temperature-dependent magnetic susceptibility measurement, one could get a useful clue on $T^>$. Due to volatile chalcogen atom, however, it is difficult to measure the physical properties of $\text{NiS}_{2-x}\text{Se}_x$ up to high temperature. On the other hand, the theoretically obtained T_{Coh} can be directly compared with that obtained from the resistivity measurement. As mentioned in the main text, the theoretically obtained T_{Coh} of NiS_1Se_1 ($x=1$) is 50 K and T_{Coh} of $\text{NiS}_{0.5}\text{Se}_{1.5}$ ($x=1.5$) is 140 K. The experimentally observed T_{Coh} of $\text{NiS}_{0.67}\text{Se}_{1.33}$ ($x=1.33$) is ~80 K.

3.) Based on Fig. R1-1 and R2-3 (b) I could imagine that the (theoretical) effective masses shown in Fig. 2 k should also carry some error bars (the eg channel in R2-3 (b) changes slightly around EF with the change in the basis set). The authors should comment on this.

As the reviewer correctly pointed out, there should be an error bar since the effective mass is evaluated from the self-energy obtained from the CTQMC impurity solver. We have evaluated the error bar originated from Monte Carlo sampling noise based on self-energy from the last 10 DMFT steps. Since we used enough Monte Carlo sampling, however, the error bar obtained from the last 10 DMFT steps is too small to be included in Fig. 2k.

Figure R1-6 a, Imaginary part of the self-energy of the NiSe₂ e_g orbital on the Matsubara axis from the last 10 DMFT steps. **b**, Doping dependent effective mass m^*/m obtained from DFT+DMFT calculation. The error bar is much smaller than the size of the triangles.

Regarding the error originated from local basis and Coulomb interaction choice mentioned by the reviewer in question 1) and 3), we cannot quantify the error rigorously. When the same interaction parameters are used in the calculation, the effective mass and the metal-insulator transition (MIT) point vary depending on the local basis and Coulomb interaction choice due to the following reasons.

- 1) The current implementation of DFT+DMFT is local basis dependent since it ignores the off-diagonal components of the hybridization matrix due to the sign problem in the impurity solver.
- 2) As we mentioned in the previous reply, full Coulomb interaction gives more itinerant results compared to Ising Coulomb interaction *when the same interaction parameters are used*. It has been reported that the MIT point varies depending on the Coulomb interaction type when the same interaction parameters are used. (See PRL 123, 236401 (2019) supplementary materials)

However, energy and temperature scale analyses should be done based on the calculation scheme (local basis, Coulomb interaction type, interaction parameters) which can describe not only the effective mass but also the phase diagram (MIT) of the system. Therefore, we believe it is not reasonable to

compare the results from different local basis or Coulomb interaction type directly based on the same interaction parameters. However, we would say the kink and temperature scale should be similar when the proper interaction parameters are used for other local basis and Coulomb interaction choices, although it should be systematically analyzed in the future study. Therefore, we did not mention the error originated from the local basis and Coulomb interaction choice.

Reply to Reviewer #2:

The authors carefully addressed my questions and suggestions in the revised manuscript. I additionally suggest that they include their response Figure R2-4 in the supplementary materials for the reader's convenience. After this minor correction I would recommend the publication of this manuscript.

We greatly appreciate the reviewer's recognition of the importance and validity of our work. We have included Figure R2-4 (comparison between ARPES and calculations) in the supplementary materials (Supplementary materials S6).

Reply to Reviewer #3:

I confirm my recommendation for this work to be published in this journal. The authors did a good job and the changes implemented meet my requests, however I did find a few things that were not fully addressed:

We thank the reviewer again for carefully reading our manuscript and for the recommendation.

1) Fig 1: The relevant compound (NiSe₂) could be mentioned in the figure

Following the reviewer's suggestion, we have modified figure 1.

2) clarification: the k-integrated spectra for panel 1d are those shown in panel 1c or 1a? the text says 1a in more than one stance, but 1a is a 2D fermi surface map, I suspect it might be a typo? if not, could you please further illustrate the integration method?

The k-integrated spectrum in panel 1d is constructed by k-integrating EDCs taken over the whole Brillouin zone. Those EDCs are also used to construct the Fermi surface map in panel 1a (1a is a plot of the spectral weight at the Fermi level). Therefore, what we meant in the manuscript is that the spectrum in 1d was obtained by k-integrating EDCs that were used to construct the Fermi surface map in 1a. We rephrase the corresponding sentences to avoid the confusion.

3) Fig. 2 caption: buried is misspelled (burried)

Thank you for pointing out the typo. It is fixed

4) figure 2 panel h: are the experimental data -from whose analysis were extracted the reported results - shown in the article or SI?

The caption only reads: "QP dispersions obtained by fitting the momentum distribution curves (MDCs)." without stating from which spectra these MDCs were taken.

Here section S5 could be referred to, for further info on the fitting, with the addition of a clarification as to what cuts were used, and if those are shown in the text.

I am guessing the analyzed spectra were 2b-e, while dopings 1.2 and 1.0 are not shown. All spectra analyzed but not shown in the main text should be reported at least in the SI.

As the reviewer correctly guessed, the dispersions in Fig. 2h are from the data in Figs. 2b – 2f but the data for $x=1.2$ and 1.0 are not shown due to the space limit. We added the data for $x=1.2$ and 1.0 to the Supplementary Materials as the reviewer suggested (as Figure S5-2).

5) figure 4: in panel a is indicated the compound (NiS₁Se₁), it would be good to do the same for panel b, since different dopings are shown. It could go near the legend.

Following the reviewer's suggestion, we have modified figure 4(b).

Of these, only point 2 and 4 are critical, while the others are more aesthetical comments.

In addition, we have slightly modified the explanation on local spin susceptibility in Calculation Details part without losing a physical meaning.